# Phosphate-binding pocket on cyclin B governs CDK substrate phosphorylation and mitotic timing

Henry Y. Ng [1], Devon H. Whelpley[1], Armin N. Adly[1], Robert A. Maxwell[2] & David O. Morgan [1] ✉

Cell cycle progression is governed by complexes of the cyclin-dependent kinases (CDKs) and their regulatory subunits cyclin and Cks1. CDKs phosphorylate hundreds of substrates, often at multiple sites. Multisite phosphorylation depends on Cks1, which binds initial priming phosphorylation sites to promote secondary phosphorylation at other sites. Here, we describe a similar role for a recently discovered phosphate-binding pocket (PP) on B-type cyclins. Mutation of the PP in Clb2, the major mitotic cyclin of budding yeast, alters bud morphology and delays the onset of anaphase. Mutation of the PP reduces multi-site phosphorylation of CDK substrates in vitro, including the Cdc16 and Cdc27 subunits of the anaphase-promoting complex/cyclosome and the Bud6 and Spa2 subunits of the polarisome. We conclude that the cyclin PP, like Cks1, controls the pattern of multisite phosphorylation on CDK substrates, thereby helping to establish the robust timing of cell-cycle events.

Complexes of cyclin-dependent kinases (CDKs) and their cyclin regulatory subunits phosphorylate hundreds of protein substrates to drive progression through the phases of the cell division cycle[1–5]. Phosphorylation generally occurs at serines or threonines followed by a proline, sometimes followed by a basic residue to form the optimal consensus site, S/T*-P-x-K/R. Multiple CDK sites are often clustered in long regions of intrinsic disorder[2]. Multisite phosphorylation in these regions alters the activity, binding partners, localization, or turnover of the regulatory and structural proteins that drive cell cycle events[6–9].

CDK substrates are phosphorylated at specific times during the cell cycle, ensuring the correct sequence and coordination of cell cycle events. The timing of substrate phosphorylation depends on many mechanisms. Due to variations in sequence context and accessibility, different CDK phosphorylation sites are phosphorylated at different rates[1–5], possibly contributing to differences in the timing of phosphorylation. Many substrates carry short linear sequence motifs (SLiMs) that interact with docking sites on a specific cyclin regulatory subunit, thereby promoting phosphorylation at cell cycle stages when that cyclin is expressed[3,10,11]. For example, an L-x-F motif on some substrates docks at a site called the hydrophobic patch of cyclin B[11]. The timing of substrate phosphorylation is also determined, at least in part, by the level of CDK activity, such that certain substrates are modified earlier in the cycle at low CDK activity and other substrates are modified later by rising levels of activity[4,5,7,8,12,13]. Phosphatase specificity for different sites also contributes to the timing of phosphorylation[14–19].

Many CDK–substrate interactions are influenced by the small accessory protein Cks1, which binds to the CDK subunit of the cyclin-CDK complex[8,13,20–25]. Cks1 has a phosphate-binding pocket with specificity for phosphorylated threonine. After initial phosphorylation of a CDK substrate at a threonine, Cks1 binds to the phosphothreonine to promote secondary phosphorylation of the substrate at downstream serines or threonines, leading to sequential multisite phosphorylation[13,24,25]. Cks1 is required for multisite phosphorylation of numerous CDK substrates. In the case of the CDK inhibitor Sic1, Cks1 promotes secondary phosphorylation at 'phosphodegron' motifs that trigger Sic1 ubiquitylation and destruction[8]. Cks1-dependent secondary site phosphorylation is thought to provide a mechanism to delay cell cycle events until CDK activity reaches a higher threshold.

[1]Department of Physiology, University of California San Francisco, San Francisco, CA, USA. [2]The Vincent J. Coates Proteomics/Mass Spectrometry Core Laboratory, University of California, Berkeley, CA, USA. ✉e-mail: David.morgan@ucsf.edu

Recent structural studies revealed that human cyclin B1 carries a binding pocket for a CDK phosphorylation site, S1126, in separase[23]. This phosphate-binding pocket (PP) is formed by the positively charged side chains of three amino acids, R307, H320, and K324, on the surface of cyclin B1 near the Cdk1 active site. These residues are highly conserved in B-type cyclins across eukaryotes (Fig. 1a and Supplementary Fig. 1a), supporting a conserved role for phosphate binding at this site. The discovery of this phosphate-binding site raised the possibility that B-type cyclins, like Cks1, interact with phosphorylation sites on Cdk1 substrates to promote secondary phosphorylation at other sites.

We showed previously that mutation of the PP in budding yeast Clb2 reduces phosphorylation of Ndd1, a transcriptional co-activator that stimulates *CLB2* expression in early mitosis[7]. Ndd1 has 16 consensus CDK phosphorylation sites. Initial phosphorylation of one site, T319, promotes Ndd1 function and *CLB2* expression in early mitosis, resulting in positive feedback[26]. When CDK activity reaches high levels in mitosis, hyperphosphorylation of Ndd1 at multiple additional sites promotes its degradation, thereby suppressing *CLB2* expression. In vitro, large numbers of sites in Ndd1 are phosphorylated by wild-type Clb2-Cdk1-Cks1 complexes. Mutation of either Cks1 or the Clb2 PP does not affect initial phosphorylation sites but reduces phosphorylation at later sites, suggesting that the cyclin PP, together with Cks1, is required for secondary phosphorylation at sites that promote Ndd1 degradation.

In budding yeast, four B-type cyclins (Clb1-4) contribute to the control of mitosis. Clb2 is the major mitotic cyclin, working with Cdk1 to catalyze phosphorylation of numerous proteins involved in mitotic processes, including spindle assembly, bud growth, and bud morphology[27–29]. It is also involved in the phosphorylation, and thus activation, of the Anaphase-Promoting Complex/Cyclosome (APC/C), the ubiquitin ligase responsible for triggering chromosome segregation in anaphase[30,31]. Deletion of *CLB2* is not lethal but causes elongated bud morphology and mitotic delays. *CLB2* is essential for survival in the absence of *CLB1* and *CLB3*[29], revealing that mitotic progression depends on contributions from multiple cyclins.

As multisite phosphorylation of CDK substrates plays an important role in cell cycle regulation, we asked whether mutation of the PP in Clb2 leads to cell cycle defects. Our results suggest that mutation of the PP causes a delay in mitotic timing, particularly when combined with deletions of *CLB1* and *CLB3*. We find that the multisite phosphorylation of several Cdk1 substrates, including subunits of the APC/C, is altered in vitro by mutation of the Clb2 PP. We propose that the precise timing of mitotic events depends on PP-dependent changes in multisite phosphorylation patterns on CDK substrates.

## Results

### The Clb2 phosphate pocket is required for normal cell proliferation

To characterize the functions and biochemical properties of the PP in B-type cyclins, we analyzed the role of the pocket in the budding yeast *Saccharomyces cerevisiae*. The three basic residues that constitute the PP of human cyclin B1 are highly conserved not only among paralogs but also orthologs from different species (Fig. 1a and Supplementary Fig. 1a). Due to its high conservation, we hypothesized that the PP plays an important role in B-type cyclin function.

Four B-type cyclins (Clb1-4) are involved in the control of mitosis in budding yeast. Clb2 is the major mitotic regulator, and we therefore focused our studies on the PP of Clb2. As described in our previous work, we constructed a strain in which *CLB2* is replaced by a mutant allele in which the three key pocket residues are mutated to alanine – termed the *clb2-pp* mutant (R366A, R379A, K383A)[7].

Mutation of the Clb2 PP did not cause apparent proliferation defects at 30 °C on rich media (Supplementary Fig. 1b; yeast strains listed in Supplementary Table 1). To examine whether other B-type cyclins compensate for the loss of the Clb2 PP, we introduced single or multiple deletion mutations in a *clb2-pp* background. In previous work, combinations of *CLB* deletions with *clb2Δ* (i.e., *clb1Δ clb2Δ, clb2Δ clb3Δ, clb1Δ clb2Δ clb3Δ, clb1Δ clb2Δ clb4Δ*, and *clb1Δ clb2Δ clb3Δ clb4Δ*) caused severe growth defects or lethality[29,32]. We found, however, that combining deletions with *clb2-pp* resulted in viable colonies (*clb1Δ clb2-pp, clb2-pp clb3Δ, clb1Δ clb2-pp clb3Δ*) (Supplementary Fig. 1b).

To provide a more detailed understanding of cell proliferation, we measured growth rate in suspension cultures using an optical density-based method. At 30 °C in rich media, all single and double mutants grew at rates that were not significantly different from that of the wild-type strain (Fig. 1b). A defect was seen only in the triple mutant, *clb1Δ clb2-pp clb3Δ*, which grew at a rate about 40% lower than that of the wild-type (Fig. 1b). Thus, in the absence of *CLB1* and *CLB3*, *clb2-pp* is not sufficient for normal rates of proliferation.

To further examine the role of the Clb2 PP, we sensitized the strains with heat stress or microtubule destabilization. The *clb1Δ clb3Δ and clb1Δ clb2-pp clb3Δ* mutants showed a mild growth defect at 37 °C (Supplementary Fig. 1b, middle). Any double or triple mutant showed a mild growth defect when treated with a low concentration of benomyl, a microtubule destabilizer (Supplementary Fig. 1b, right).

Analysis by microscopy revealed that cells of the *clb2-pp* strain do not display an elongated morphology phenotype like that seen in a *clb2Δ* strain[29,32] (Fig. 1c). *clb2-pp clb3Δ* cells showed a mild elongated morphology, while *clb1Δ clb2-pp* and *clb1Δ clb2-pp clb3Δ* cells displayed a more pronounced elongated phenotype accompanied by cell chains (Fig. 1c). Thus, the phosphate pocket is required for the normal function of Clb2 in bud morphology.

### The Clb2 phosphate pocket is required for normal mitotic progression

We next investigated the role of the Clb2 PP in progression through mitosis. As in our previous work, we measured mitotic progression in single cells by analyzing the behavior of a fluorescently tagged spindle pole body (SPB)[33]. The SPB is duplicated in early S phase and separates slightly at the onset of mitosis to form the short mitotic spindle; the initiation of anaphase then triggers rapid SPB separation as the spindle elongates across the mother cell and into the bud[34,35]. The time between initial SPB separation and spindle elongation reflects the time from mitotic entry to anaphase onset.

We constructed wild-type and *clb2-pp* strains in which the SPB protein Spc42 was fused to a C-terminal mCherry fluorophore, and we measured the time from SPB separation to spindle elongation (Fig. 2a and Supplementary Fig. 2a). As in our previous studies, this time was highly variable in wild-type cells, ranging from 10 to 66 min with a median of 32 min[33]. In comparison, *clb2-pp* mutant cells displayed a range of 20 to 82 min, with a median of 48 min. Thus, *clb2-pp* cells experience a ~15 min delay in a ~30 min process (Fig. 2a). We conclude that the Clb2 PP is required for the normal timing of progression through mitosis to anaphase onset.

We next analyzed mitotic progression by western blotting of mitotic proteins in cells released from a G1 arrest. We first analyzed the separase inhibitor securin/Pds1, which is known to increase in abundance in S phase and then decline rapidly at the onset of anaphase due to its ubiquitylation by the Cdc20-activated APC/C. Mutation of the Clb2 PP resulted in a small but reproducible delay in the timing of securin destruction (Fig. 2b, left). Deletion of *CLB1* and *CLB3* had no significant impact on securin degradation, but *clb1Δ clb2-pp clb3Δ* cells showed a major delay in securin degradation past 120 min after release from G1 (Fig. 2b, right). Together with the mitotic timing results above, these results suggest that the Clb2 PP is required for the normal timing of securin destruction and thus the timing of sister-chromatid separation in anaphase.

We also monitored the degradation timing of the transcriptional co-activator Ndd1. Ndd1 levels are regulated by multiple mechanisms.

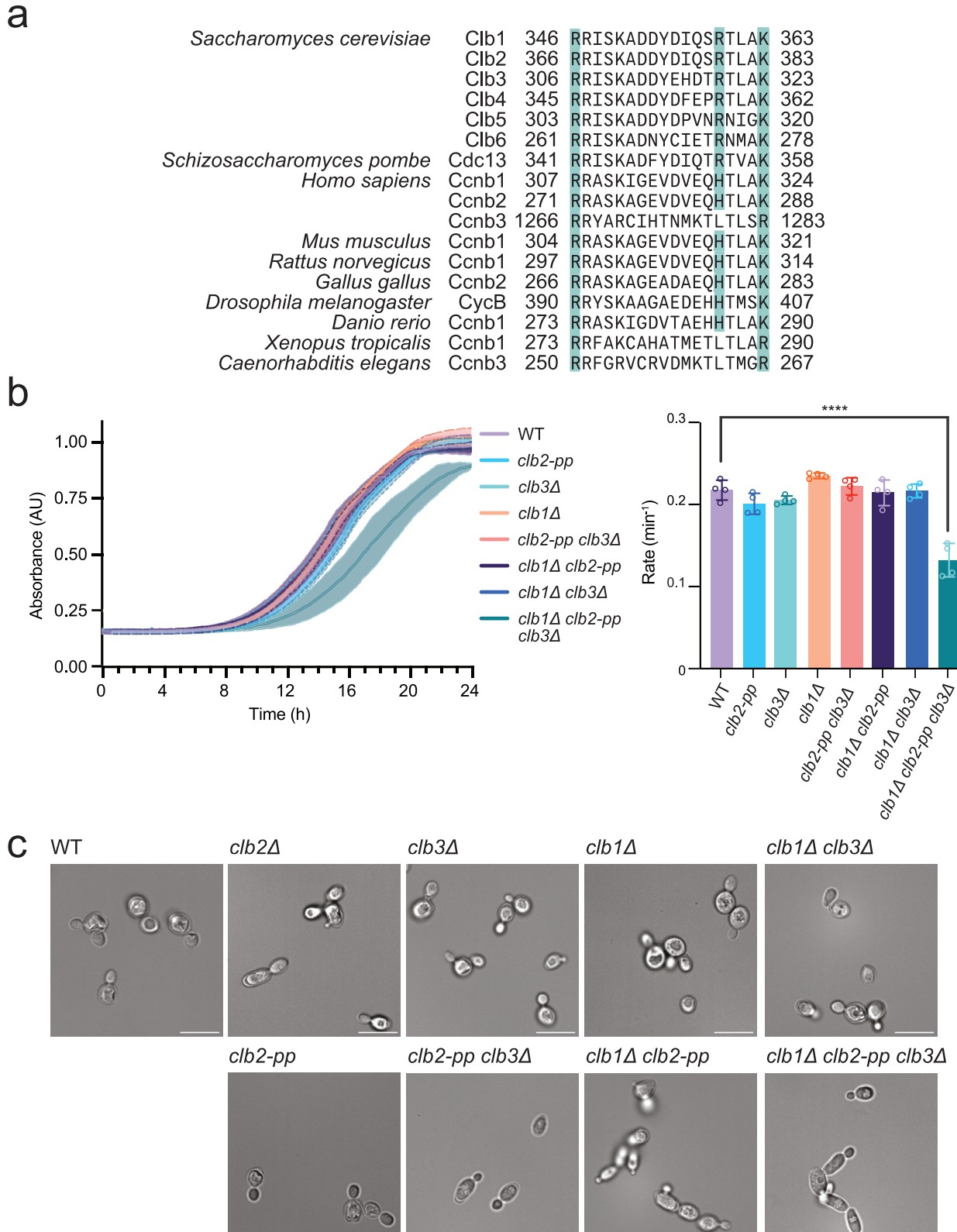

**Fig. 1 | Mutation of the conserved phosphate-binding pocket in Clb2 causes a growth defect in the absence of *CLB1* and *CLB3*. a** Aligned amino acid sequences of B-type cyclins, with cyan indicating the three basic residues that constitute the phosphate-binding pocket. These residues are not conserved in A-type cyclins. **b** (left) Growth curves of indicated yeast strains, with $OD_{600}$ recorded every 15 min. Data represent mean ± standard deviation (SD) of four independent biological replicates. (right) Growth rates calculated by fitting $OD_{600}$/min to a logistic growth model. Data represent mean ± SD of four independent biological replicates. Statistical significance was determined using one-way analysis of variance (ANOVA) (****$p < 0.0001$). **c** Representative differential interference contrast (DIC) microscopy images of the indicated yeast strains at mid-log phase in rich (YPD) media at 30 °C. Scale bar is 10 µm. Source data are provided as a Source Data file.

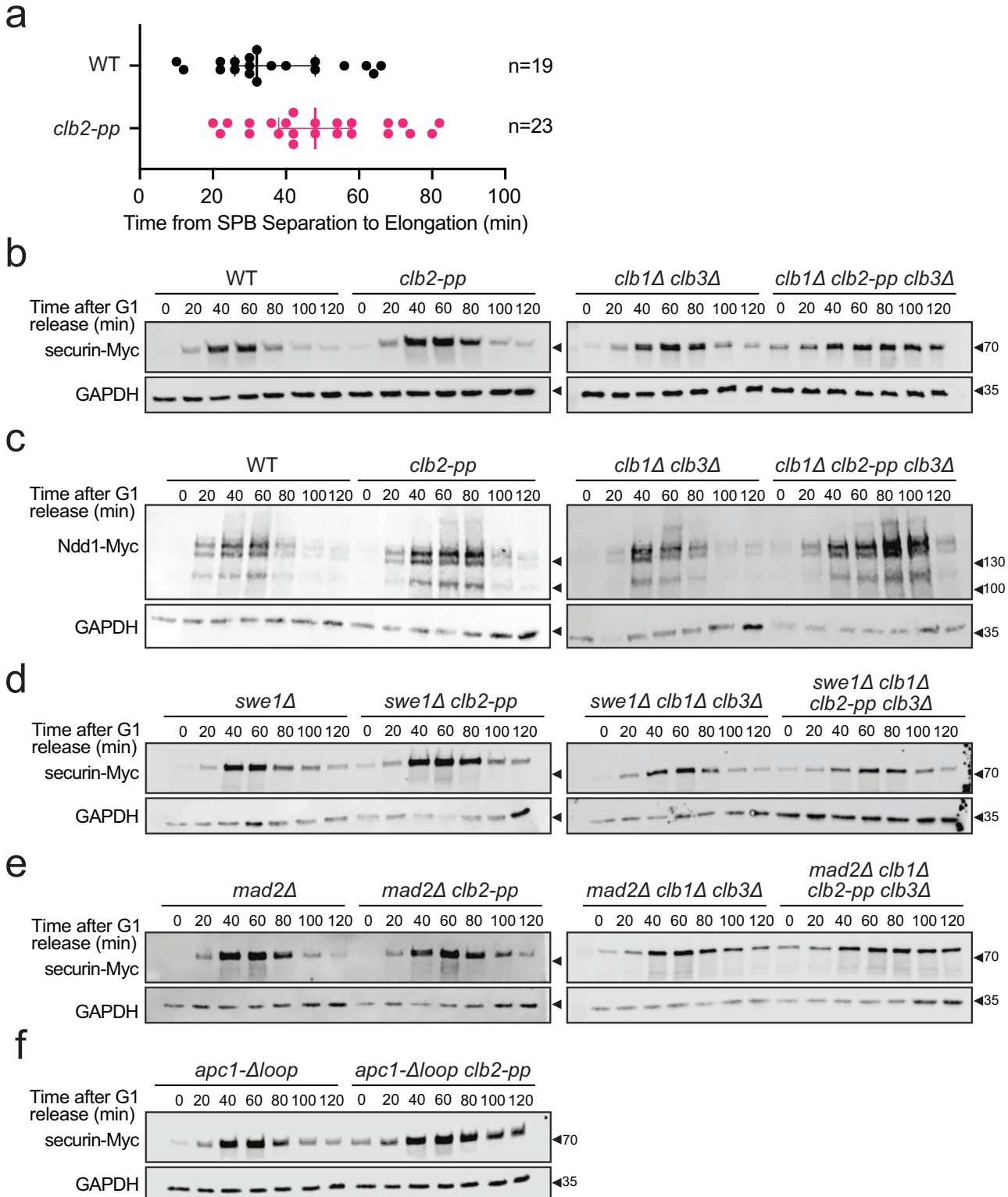

**Fig. 2 | The *clb2-pp* mutation causes a mitotic delay. a** Single-cell measurements of time from SPB separation to spindle elongation in asynchronous wild-type (WT) or *clb2-pp* cells carrying mCherry-labeled Spc42. Each data point represents a single cell, and the bars indicate median ± SD. The sample size (n) is noted at right. See Supplementary Fig. 2a for representative images. **b** Western blots of Myc-tagged securin after release from a G1 arrest in the indicated yeast strains. Alpha factor was added back at the 60 min time point. GAPDH loading control on the lower blot. Representative of 10 independent experiments. **c** Western blots of Myc-tagged Ndd1 after release from a G1 arrest in the indicated yeast strains. These strains carry a second copy of Ndd1-Myc under the control of a *GAL* promoter, and experiments were performed in galactose media. Alpha factor was added back at the 60 min time point. GAPDH loading control on the lower blot. Representative of 2 independent experiments. **d** Same as (**b**) with *swe1Δ*. Representative of 2 independent experiments. **e** Same as (**b**) with *mad2Δ*. Representative of 2 independent experiments. **f** Same as (**b**) with *apc1-Δloop*. Representative of 2 independent experiments. Source data are provided as a Source Data file.

We showed previously that high Clb2-Cdk1 levels drive PP-dependent multisite phosphorylation of Ndd1, triggering its destruction in a prolonged spindle assembly checkpoint arrest; mutation of the Clb2 PP therefore stabilizes the protein prior to anaphase[7]. In anaphase, Ndd1 is then degraded via the Cdh1-activated form of the APC/C[36]. Here, western blotting of Ndd1 levels in cells released from a G1 arrest showed that the anaphase degradation of Ndd1, like that of securin, was delayed 10–20 min in the *clb2-pp* strain, and more severely delayed in *clb1Δ clb2-pp clb3Δ* cells (Fig. 2c). These results are consistent with a delay in the onset of anaphase in *clb2-pp* cells.

Anaphase onset depends on Clb2-Cdk1-mediated multisite phosphorylation of several APC/C subunits, triggering APC/C activation by the activator subunit Cdc20[30,31,37]. A simple explanation for our results is that the Clb2 PP promotes APC/C-Cdc20 activation. We considered three possible mechanisms by which this might occur: (1) the *clb2-pp* mutation causes cytoskeletal defects that activate the morphogenesis checkpoint, resulting in suppression of Cdk1 activity by the inhibitory kinase Swe1[37]; (2) the *clb2-pp* mutation causes mitotic spindle defects that lead to APC/C inhibition by the spindle assembly checkpoint (SAC)[38,39]; or (3) the Clb2 PP is directly required for multisite phosphorylation of APC/C subunits by Cdk1[30,31].

To test whether the mitotic delay in the *clb2-pp* mutant is caused by activation of the morphogenesis checkpoint, we monitored securin degradation in strains lacking Swe1. In comparison to *swe1Δ* cells, *swe1Δ clb2-pp* cells displayed a mild delay in securin destruction, similar to that in *clb2-pp* cells (Fig. 2d), suggesting that the small mitotic delay in *clb2-pp* cells is not due to Cdk1 inhibition by Swe1. However, deletion of *SWE1* did cause a partial reduction in the prolonged mitotic delay in *clb1Δ clb2-pp clb3Δ* cells, suggesting that this delay results in part from Cdk1 inhibition by Swe1, consistent with the bud morphology defect we observe in these cells. To test if the *clb2-pp* anaphase delay is caused by activation of the SAC, we analyzed securin levels in strains lacking Mad2, a critical SAC component[38,39]. Deletion of *MAD2* did not reduce the minor mitotic delay in *clb2-pp* cells or the lengthy delay in *clb1Δ clb2-pp clb3Δ* cells (Fig. 2e), suggesting that SAC activation does not occur in these cells.

Consistent with these results, deletion of *SWE1* partially rescued the growth defect of *clb1Δ clb2-pp clb3Δ* cells, but deletion of *MAD2* did not (Supplementary Fig. 2b).

We next sought a genetic approach to test the possibility that the *clb2-pp* mutation directly causes defects in Cdk1-mediated APC/C phosphorylation. An ideal approach would be to test if the *clb2-pp* phenotype is rescued by an APC/C mutant that is active in the absence of phosphorylation. Such a mutant is known to be possible with the vertebrate APC/C[40–43]. In vertebrate cells, an autoinhibitory loop in the Apc1 subunit blocks Cdc20 binding in the absence of phosphorylation. Phosphorylation of other APC/C subunits (primarily Cdc27/Apc3) is thought to recruit Cdk1-cyclin B1-Cks1 via binding of Cks1 to phosphorylated residues, which leads to secondary phosphorylation of multiple sites in the autoinhibitory loop of Apc1 – thereby displacing the loop and enabling Cdc20 binding. Deletion of the autoinhibitory loop of human Apc1 results in phosphorylation-independent Cdc20 binding[40,42].

It is not known if similar mechanisms activate the budding yeast APC/C. Yeast Apc1 contains a disordered loop (aa 225–365) at the same location as the autoinhibitory loop of human Apc1, but the sequences are not related. The yeast loop contains only one Cdk1 consensus site as compared to the seven sites in the human loop. Recent structural studies suggest that the Apc1 loop of budding yeast APC/C does not occupy the Cdc20 binding site as it does in human APC/C[44]. Nevertheless, we tested the possibility that deletion of this loop from yeast Apc1 rescues the anaphase defects in *clb2-pp* cells.

We constructed the *apc1-Δloop* strain, in which the *APC1* gene encodes an Apc1 protein lacking aa 225–365. This strain displayed a minor growth defect on rich media plates at 30 °C or 37 °C, and on plates containing the microtubule drug benomyl (Supplementary Fig. 3a). A minor growth defect was observed in cells grown in suspension in rich media at 30 °C (Supplementary Fig. 3b). The timing of securin destruction in *apc1-Δloop* cells was normal (Fig. 2f). The double mutant (*apc1-Δloop clb2-pp*) displayed a more severe growth phenotype on plates and in suspension (Supplementary Fig. 3a, b), as well as a severe delay in the timing of securin destruction (Fig. 2f). Together, these results suggest that deletion of the Apc1 loop does not result in APC/C hyperactivation that rescues the *clb2-pp* defect; on the contrary, it seems more likely that loop deletion does not bypass the requirement for phosphorylation but simply reduces APC/C function. In this case, the synthetic defect in the double mutant is consistent with a role for the PP in APC/C function.

### The Clb2 phosphate pocket accelerates APC/C subunit phosphorylation in vitro

We turned to biochemical approaches to determine the impact of the PP on phosphorylation of Cdk1 substrates in vitro. For these experiments, we produced large amounts of homogeneous *S. cerevisiae* Clb2-Cdk1-Cks1 complexes by a multi-step procedure: yeast Cdk1 and Cak1 were expressed separately from baculoviruses, purified, and mixed to generate T169-phosphorylated Cdk1[45]; Clb2 (wild-type or the Clb2-pp mutant) was expressed in bacteria, purified, and mixed with phosphorylated Cdk1 to generate active Clb2-Cdk1 complexes; and yeast Cks1 (wild-type or mutated at three key residues in its phosphothreonine-binding pocket)[8,20] was expressed in bacteria, purified, and mixed with Clb2-Cdk1.

As discussed earlier, a likely explanation for the anaphase defect in *clb2-pp* cells is that the phosphate pocket is required for full phosphorylation of the APC/C. Several APC/C subunits, including Cdc16, Cdc23, Cdc27, and Apc9, contain disordered regions with multiple CDK consensus sites that are phosphorylated in mitosis and by Clb2-Cdk1-Cks1 in vitro[30,31,37,42]. Most Cdk1 consensus sites in the APC/C are suboptimal sites whose phosphorylation requires high levels of Clb2-Cdk1 activity working in concert with the Cks1 subunit.

To assess the importance of the Clb2 PP in APC/C phosphorylation, APC/C was immunopurified from lysates of a *CDC16-TAP* yeast strain and incubated with purified Clb2-Cdk1-Cks1 and radiolabeled ATP (Fig. 3a). Phosphorylation occurred primarily on the same three subunits seen in previous studies of APC/C phosphorylation in vitro: Cdc16, Cdc27, and Apc9[30,31,37], with minor phosphorylation of Cdc23. Mutation of the Clb2 PP reduced the rate of phosphorylation of all subunits, indicating that the phosphate-binding pocket promotes phosphorylation at multiple sites.

To understand APC/C subunit phosphorylation in more detail, we analyzed the phosphorylation of bacterially-expressed fragments encompassing the major disordered regions of Cdc16 (aa 31–180) or Cdc27 (aa 241–360), which contain most of the Cdk1 consensus sites in these proteins (Supplementary Fig. 4; bacterial expression constructs listed in Supplementary Table 2). To estimate the stoichiometry of phosphorylation, radiolabeled reaction products were separated by Phos-tag SDS-PAGE.

In pilot experiments, we compared Cdc16 phosphorylation patterns with wild-type Clb2-Cdk1 bound to either wild-type or mutant Cks1 (Supplementary Fig. 5a). As expected for Cks1-dependent priming, initial phosphorylation rates were similar for wild-type and mutant Cks1, but the wild-type Cks1 greatly enhanced later phosphorylation at multiple secondary sites. Interestingly, basal kinase activity was reduced in reactions lacking any Cks1, suggesting that Cks1 enhances general kinase activity or stability even when lacking its phosphate-binding site.

We next compared Cdc16 phosphorylation patterns in reactions with wild-type Clb2 and the Clb2-pp mutant. Initial phosphorylation rates were similar in all reactions, but mutation of the Clb2 PP caused a moderate reduction in multisite secondary phosphorylation,

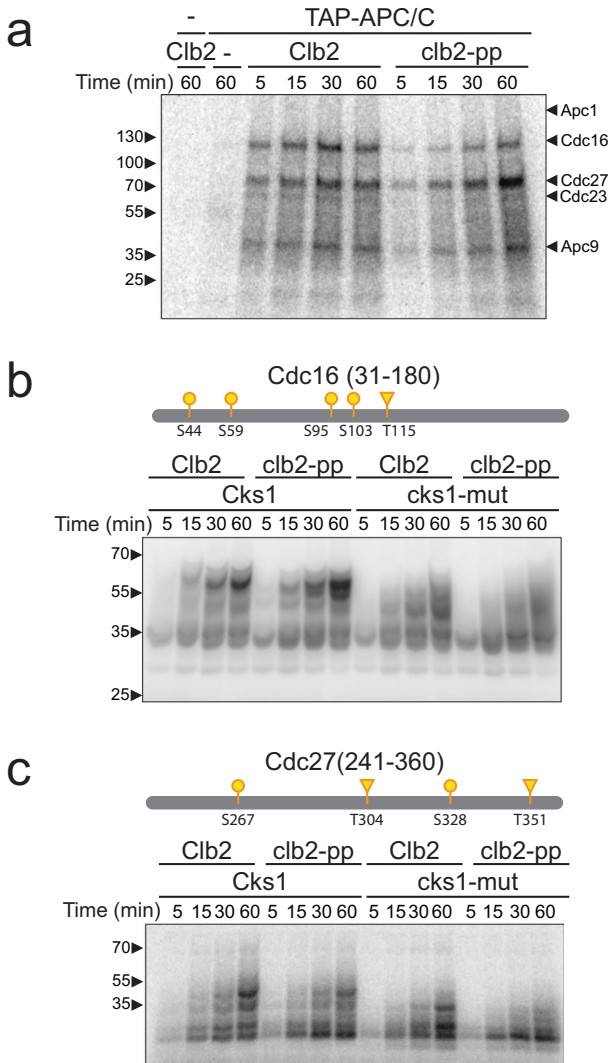

**Fig. 3 | Phospho-pocket mutations reduce phosphorylation of APC/C subunits in vitro. a** APC/C was immunoprecipitated from yeast lysate and treated with wild-type or *clb2-pp* Clb2-Cdk1 plus mutant Cks1 and radiolabeled ATP. Reaction products were analyzed by SDS-PAGE and autoradiography (lane 1: kinase only control; lane 2: no kinase control; lane 3-6: wild-type Clb2; lane 7-10: Clb2-pp). APC/C subunits were identified based on previous studies[37]. **b** 2.5 μM purified Cdc16 fragment (aa 31–180) was incubated with 150 nM wild-type or *clb2-pp* Clb2-Cdk1 plus wild-type or mutant Cks1 and radiolabeled ATP. Diagrams at the top indicate suboptimal (S/T*-P; yellow) CDK consensus sites (S: circle; T: triangle) in the fragment (see Supplementary Fig. 4 for complete sequences). Representative of 10 independent experiments. **c** Same as (**b**) with 2.5 μM purified Cdc27 fragment (aa 241–360) and 100 nM Clb2-Cdk1. Representative of 8 independent experiments. Coomassie Blue-stained gels for (**b** and **c**) are found in Supplementary Fig. 6. Source data are provided as a Source Data file.

particularly in the presence of the mutant Cks1 (Fig. 3b and Supplementary Fig. 6a).

These results are consistent with the model that the PP (like Cks1) interacts with a priming phosphorylation site, resulting in a higher affinity (lower $K_M$) than that for an unprimed substrate. According to this model, the effect of the PP (or Cks1) should depend on the balance of kinase and substrate concentrations – the PP will have less effect if substrate concentration is high and/or kinase activity is insufficient to modify a significant fraction of the substrate population; under such conditions the unmodified substrate would outcompete the small amount of phosphorylated substrate. Consistent with this model, we

found that multisite phosphorylation and the effect of PP mutations were abolished when the substrate was in great excess over the kinase (Supplementary Fig. 5b). Thus, the impact of the PP is most apparent at moderate substrate and kinase concentrations that lead to modification of a significant fraction of the substrate.

We next analyzed phosphorylation of the Cdc27 fragment. In the presence of wild-type Cks1, wild-type Clb2-Cdk1 catalyzed the phosphorylation of 4-5 sites, roughly equal to the four suboptimal Cdk1 consensus sites on the protein (Fig. 3c and Supplementary Fig. 6b). Mutation of Cks1 reduced multisite phosphorylation, while mutation of the PP had a less dramatic effect that was seen primarily in reactions with mutant Cks1. In these and other Cdc27 studies below, the effect of the PP was most apparent in the relative amounts of the two lowest bands on the autoradiograph, which are likely to reflect mono- and di-phosphorylated species: in reactions with wild-type Clb2 the lower band shifts to the second band, while in reactions with Clb2-pp the lower band remains dominant.

The Cdc27 fragment contains four suboptimal CDK consensus sites, with proline at the +1 position but no basic residue at + 3 (Supplementary Fig. 4). Mutation of all four sites abolished phosphorylation by Clb2-Cdk1 (Fig. 4a and Supplementary Fig. 7a). We then analyzed 'single-site' mutants carrying just one of the four CDK consensus sites. Analysis of these mutants revealed that the two serines, S267 and S328, are more rapidly phosphorylated than the two threonines, T304 and T351 (Fig. 4a and Supplementary Fig. 7a).

Past work suggests that Cks1-mediated secondary phosphorylation can occur at non-CDK consensus sites (i.e., sites lacking the proline after the S or T). Interestingly, in no single-site mutant did we observe significant lower-mobility bands above the mono-phosphorylated species. Thus, if any of the four sites alone can act as a priming site, then priming drives secondary phosphorylation mostly at other CDK consensus sites.

We next analyzed a mutant in which the two serines were mutated to alanines (S267A, S328A), leaving intact the two threonines. Consistent with the poor phosphorylation of these threonines in the single-site mutants, the double serine mutant displayed a low level of phosphorylation in two bands, and there was no clear effect in the PP mutant (Fig. 4b and Supplementary Fig. 7b). Thus, one or both of the serine residues are required for the effects of the PP, either by acting as priming sites or as secondary sites.

Mutation of single sites in Cdc27 to alanine revealed that no single site is critical for PP-mediated effects. Mutation of either serine alone (S267A or S328A) had mild inhibitory effects; in both cases the putative di-phosphorylated species accumulated rapidly in the wild-type Clb2 reactions but not in reactions with the PP mutant (Fig. 4b and Supplementary Fig. 7b). Thus, both serines might serve as priming and/or secondary phosphorylation sites.

The T304A mutant, unlike the other mutants, reduced the appearance of the di-phosphorylated species, resulting in accumulation of the mono-phosphorylated lower band in reactions with wild-type Clb2 (Fig. 4b and Supplementary Fig. 7b). Mutation of the PP had minor but reproducible effects on the pattern of phosphorylated species. Mutation of T351 had mild effects on the pattern of phosphorylation, similar to the effects of the serine mutations (Fig. 4b and Supplementary Fig. 7b). Together, our results with Cdc27 mutants do not provide definitive evidence that any single site is a critical priming or secondary site.

### The Clb2 phosphate pocket promotes phosphorylation of polarisome subunits

We chose two additional Cdk1 substrate candidates for analysis based on the budding defects observed in *clb2-pp* strains. We tested subunits of the polarisome complex, which serves as an actin-organizing center for polarized bud growth[46]. Two polarisome subunits, Bud6 and Spa2, are known to be Cdk1 targets involved in the regulation of

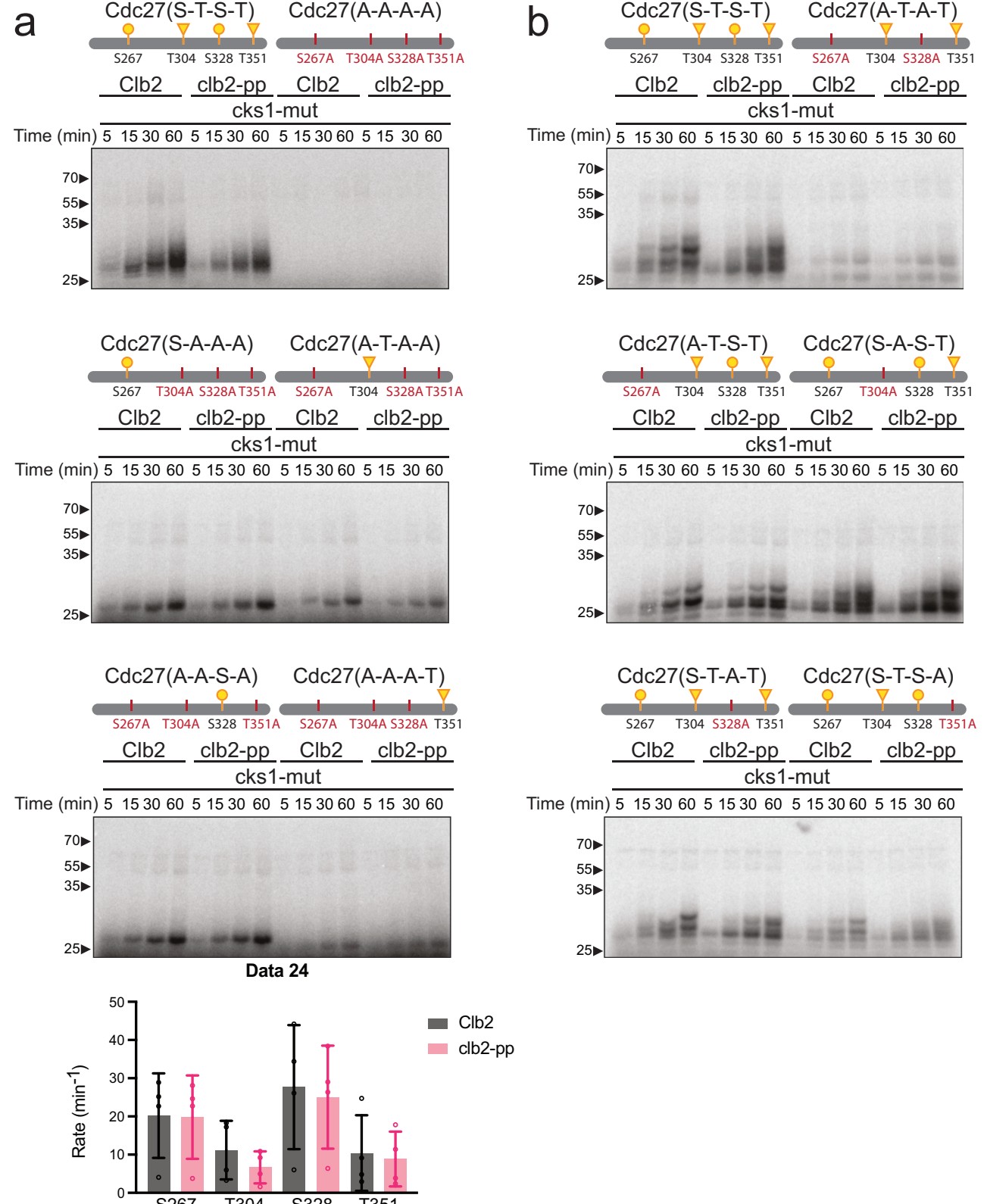

**Fig. 4 | Phospho-pocket mutations reduce phosphorylation of CDK consensus sites on Cdc27. a**, **b**, 2.5 μM of the indicated Cdc27 mutant fragments were incubated with 100 nM wild-type or *clb2-pp* Clb2-Cdk1 plus mutant Cks1 and radiolabeled ATP. Reaction products were analyzed by Phos-tag SDS-PAGE and autoradiography. Diagrams at the top indicate suboptimal (S/T*-P; yellow) CDK consensus sites (S: circle; T: triangle) in the tested fragment (see Supplementary

Fig. 4 for complete sequences). Representative of 4 independent experiments. Coomassie Blue-stained gels in Supplementary Fig. 7. For the graph at the bottom of panel (**a**), phosphate incorporation into the single-site mutants over time was used to calculate the mean phosphorylation rate (± SD) in 4 independent experiments. Source data are provided as a Source Data file.

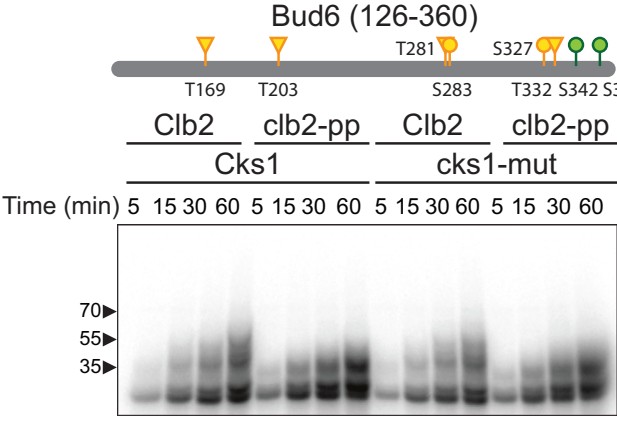

**Fig. 5 | Phospho-pocket mutations reduce phosphorylation of polarisome subunits.** 2.5 μM of purified Bud6 fragment (aa 126–360) or 5 μM of purified Spa2 fragment (aa 524–670) was incubated with 100 nM wild-type or *clb2-pp* Clb2-Cdk1 plus wild-type or mutant Cks1 and radiolabeled ATP. Diagrams at top indicate optimal (S/T*-P-x-K/R; green) and suboptimal (S/T*-P; yellow) CDK consensus sites (S: circle; T: triangle) in the tested fragment (see Supplementary Fig. 4 for complete sequences). Representative of 3 independent experiments. Coomassie Blue-stained gels in Supplementary Fig. 8. Source data are provided as a Source Data file.

budding[1,46,47]. We examined phosphorylation of disordered protein fragments from Bud6 (aa 126–360) and Spa2 (aa 524–670), both of which contain numerous suboptimal and optimal Cdk1 consensus sites (Supplementary Fig. 4).

Multisite phosphorylation of both Bud6 and Spa2 was inhibited by mutation of the Clb2 PP (Fig. 5 and Supplementary Fig. 8). Mutation of the PP had a particularly striking effect on Spa2 phosphorylation in reactions with wild-type or mutant Cks1 – unlike the other substrates, where Cks1 effects dominate. The impact of the PP on Spa2 is illustrated not just in the autoradiographs of radiolabeled reaction products (Fig. 5) but also in the Coomassie Blue-stained gels (Supplementary Fig. 8). Unmodified substrate is depleted in reactions with the PP mutant, as expected if the mutant kinase has no preference for modified substrate.

### The Clb2 phosphate pocket alters phosphorylation at specific sites on Cdk1 substrates

We used label-free quantitative mass spectrometry (MS) to identify and quantify phosphorylation sites in our kinase reactions. We carried out large-scale reactions with the four substrate fragments as described above, but without radiolabeled ATP. We also included the Cdc27 mutant carrying only S328 and lacking the other three CDK consensus sites. Radiolabeled reactions were carried out in parallel and analyzed by Phos-tag SDS-PAGE to confirm extensive multisite phosphorylation after a 60-min reaction (Supplementary Fig. 9). Aliquots of each unlabeled 60 min reaction were trypsinized, and the resulting peptides were subjected to liquid chromatography and tandem MS for identification of phosphorylation sites. Seven aliquots of each reaction were analyzed separately to provide technical replicates.

Accurate phosphorylation site assignment was difficult due to the absence of quality tryptic cleavage sites (resulting in long peptides) and because all substrates contain regions where multiple uncertain phosphate assignments were clustered on single peptides (the substrate fragments contain ~24–29% serines and threonines) (Supplementary Data 1). We used ion intensity and AScore thresholds to filter out most low-confidence phosphopeptides. We increased stringency further by selecting phosphopeptides with measurable chromatographic peak areas in four or more replicates. Normalized peak areas were used to compare the amount of phosphorylation at specific sites by wild-type and PP mutant kinases (Supplementary Data 2 and Fig. 6).

CDK consensus sites were the major sites of phosphorylation in all substrates, and the optimal CDK sites of Bud6 and Spa2 were particularly well modified. In addition, multiple sites were identified that lack the +1 proline of the CDK consensus. Of the 19 non-consensus sites we observed, 11 include a basic residue at +3 or +4 and might therefore be considered as partial consensus sites.

The impact of PP mutation varied among the different substrates and sites (Fig. 6). In Cdc16, the PP mutation had little effect on highly modified CDK consensus sites but caused a decrease in phosphorylation at an N-terminal consensus site, S44, and a non-consensus site, T76. Similarly, most CDK consensus sites in Bud6 were unaffected by the PP mutation, but several non-consensus sites exhibited reduced phosphorylation in the PP mutant (S143, T155, S172, T212, T240, T246, T280, S285). In Cdc27, phosphorylation of three CDK consensus sites was not affected by the PP mutation, but phosphorylation of the fourth CDK site, S267, was reduced. In the analysis of Cdc27 carrying only S328 and not the other consensus sites, a small amount of non-consensus phosphorylation was observed at T292 with wild-type kinase and was reduced in the PP mutant, consistent with our evidence that this protein is primarily mono-phosphorylated on Phos-tag gels (Fig. 4a).

Surprisingly, mutation of the PP did not cause significant changes in phosphorylation sites in Spa2 (Fig. 6), even though this substrate displays clear PP-dependent effects on Phos-tag gels (Fig. 5 and Supplementary Fig. 9). One possible explanation is that MS analysis centers on abundant sites that meet stringent selection criteria, and the PP might have greater effects on less readily detected sites. Spa2, for example, contains two long tryptic peptides with large numbers of sites that cannot be assigned with confidence and are therefore missing from the analysis in Fig. 6. The other substrates also contain such regions. Furthermore, because MS analysis focuses primarily on single sites in tryptic peptides, it cannot provide much information about the number of sites simultaneously modified across the entire protein, as seen on Phos-Tag gels. Multiple sites might be distributed across many proteins in the PP mutant reactions, but collected on fewer proteins with the wild-type kinase. Further studies will be required to unravel the complex patterns of multisite phosphorylation that depend on the PP and other docking sites.

### Discussion

Our results shed light on the biological function of the highly conserved phosphate-binding pocket of mitotic B-type cyclins. We found that mutation of the PP in budding yeast Clb2 had little impact on cell morphology or proliferation rate under normal conditions, but detailed analyses uncovered defects in the timing of anaphase onset. More pronounced defects in morphology, proliferation, and mitosis were observed when the *clb2-pp* mutation was combined with deletions of the two other major mitotic cyclin genes, *CLB1* and *CLB3*. We

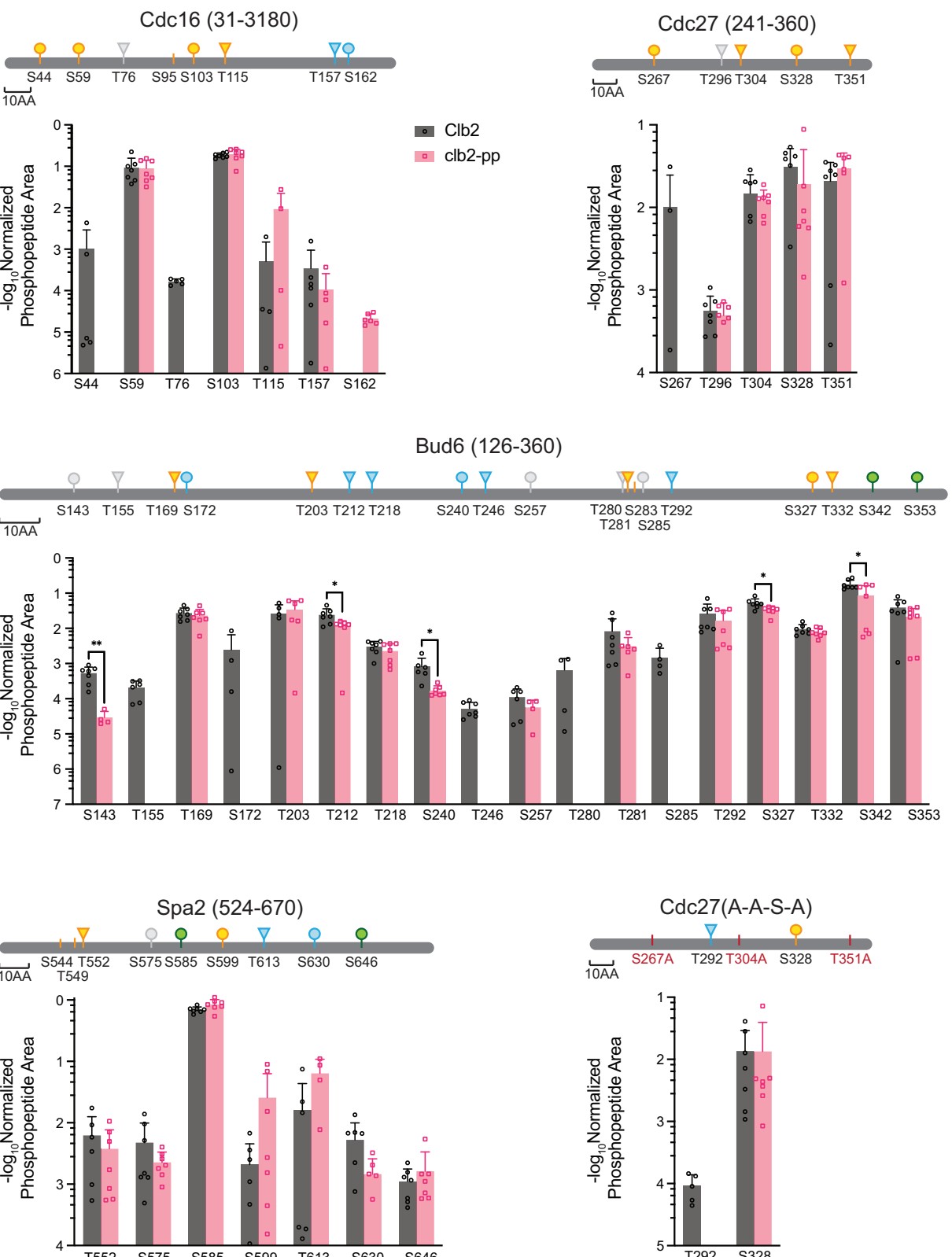

**Fig. 6 | Quantification of phosphorylation at specific sites on CDK substrates.** The indicated five substrates (10 μM) were incubated for 60 min in 55 μl reaction volumes containing 200 nM wild-type or *clb2-pp* Clb2-Cdk1 plus 200 nM mutant Cks1. Samples were trypsinized, and seven aliquots from each reaction were subjected to LC-MS/MS. High-confidence phosphopeptides (2% ion intensity, AScore 15) were tabulated with the PTM Profiles function in PEAKS software, which provides the chromatographic peak areas for modified peptides. To normalize for the quantity of substrate, modified peptide peak areas were divided by the total peak area of peptides from the Smt3 tag at the amino-terminus of the substrate.

Normalized quantities are plotted here (mean +/− SD; *n* = 4–7 replicates; see Supplementary Data 2). Asterisks indicate results of two-tailed unpaired t-tests (*$p < 0.05$; **$p < 0.01$). Diagrams at the top indicate optimal (S/T*-P-x-K/R; green) and suboptimal (S/T*-P; yellow) CDK consensus sites. Blue indicates non-consensus sites lacking a proline at +1 but containing a basic (K/R) residue at +3 or +4; other non-consensus sites are colored gray (see Supplementary Fig. 4 for complete sequences). A small number of CDK consensus sites were not confidently quantified and are shown as vertical marks (S95 in Cdc16, S283 in Bud6, S544 and T549 in Spa2). Source data are provided in Supplementary Data 2 and the Source Data file.

conclude that the PP is required for normal mitotic progression and potentially for other Clb2 functions, and that other cyclins provide some of these functions in the absence of the Clb2 PP.

The *clb2-pp* mutation, when combined with deletions of *CLB1* and *CLB3*, causes an elongated morphology that resembles the well-known bud morphology defect of *clb2Δ* cells, in which there is a delay in the switch from polar to isotropic bud growth[27,29,32]. Our results suggest that PP-dependent Clb2-Cdk1 substrates function in this switch. For example, Clb2-Cdk1 is thought to phosphorylate components of the polarisome, a protein complex that controls actin dynamics and polarized bud growth[46]. Potential Clb2-Cdk1 substrates in this complex include Spa2 and Bud6, both of which are extensively phosphorylated on multiple Cdk1 consensus sites and other sites[1,46,47]. We showed that multisite phosphorylation of Spa2 and Bud6 are PP-dependent, raising the possibility that the bud morphology defect in *clb2-pp* cells is due in part to reduced phosphorylation of these proteins.

Our evidence that the Clb2 PP is required for timely anaphase onset is consistent with abundant previous evidence that Cdk1 and mitotic cyclins play important roles in the metaphase-anaphase transition. Delays in anaphase onset are seen in cells carrying defects in Cdk1 activity or cyclin expression, including cells lacking *CLB2*[30,31,37,48,49]. The temperature-sensitive *clb2-IV* mutant (D232G, L286S, K353R, D485G) exhibits delayed securin degradation when combined with a *CLB1* deletion[48]. In principle, Clb2-dependent mitotic delays might be expected to result from defects in early mitotic processes such as spindle assembly, which would delay anaphase by activation of the SAC[38,39]. However, our data suggest that the anaphase delay of the *clb2-pp* mutant is independent of the SAC.

Cdk1-dependent phosphorylation of multiple APC/C subunits is known to promote APC/C binding to the activator subunit Cdc20, thereby initiating the ubiquitylation and destruction of securin and mitotic cyclins[30,31,40–43]. APC/C activation depends on multisite phosphorylation of disordered loops in conserved subunits (Cdc27, Cdc16, and Cdc23 in budding yeast). The Cks1 subunit is important for Cdc16 and Cdc27 hyperphosphorylation in yeast, and APC/C activation depends on Cks1-mediated APC/C hyperphosphorylation in vertebrates[30]. In this study, we found that phosphorylation of multiple subunits in the intact yeast APC/C is promoted by the phosphate pocket, and we also observed effects of the PP mutation on the phosphorylation of Cdc16 and Cdc27 fragments. These results support the possibility that the anaphase defect in *clb2-pp* cells is due to reduced phosphorylation of APC/C subunits. We also observed that deletion of *SWE1* partially reduced the anaphase defect in *clb1Δ clb2-pp clb3Δ* cells, suggesting that Cdk1 catalytic activity is reduced by Swe1 in these cells, further inhibiting APC/C phosphorylation.

In vertebrates, Cks1-dependent binding of Cdk1 to APC/C subunit loops is thought to promote secondary phosphorylation on the auto-inhibitory loop in the large Apc1 subunit, leading to the formation of a binding site for Cdc20[30,31,37,40,42,43,50]. However, the mechanism of APC/C activation by phosphorylation in yeast remains unclear. The sequence of the putative autoinhibitory loop of yeast Apc1 is not related to the vertebrate sequence and does not contain abundant CDK consensus sites. In a recent structural analysis of yeast APC/C, the activator binding site is not occupied by the Apc1 loop as it is in the human complex[44]. Our evidence does not support a role for this loop in autoinhibition. On the contrary, we found that deletion of the loop in yeast Apc1 causes a growth defect that is more severe in combination with the *clb2-pp* mutation, suggesting that deletion of the loop reduces APC/C function. Further studies will be required to decipher the mechanism by which yeast APC/C phosphorylation promotes Cdc20 binding.

Given the large number of Cdk1 substrates carrying multiple clustered phosphorylation sites, the *clb2-pp* phenotype is likely to be complex and arise from changes in the phosphorylation state of many proteins. For example, the high levels of Ndd1 that result from hypophosphorylation[7] would be expected to promote changes in the expression of multiple mitotic genes. Numerous other proteins are likely to be underphosphorylated, resulting in a variety of subtle defects that are not apparent in simple studies of cell cycle progression in rich laboratory conditions. Furthermore, it is possible that the PP is not simply involved in the enhancement of CDK substrate phosphorylation; it might also have roles in regulatory interactions. Indeed, the pocket was initially identified through its critical role in the inhibitory interaction between human Cdk1-cyclin B1 and phosphorylated separase[23]. Mutations in the phosphate pocket were found many years ago to inhibit interactions with the CDK regulator Cdc25[51]. We therefore expect that PP mutations have a broad impact on CDK function and regulation in the cell cycle.

The simple model of PP function is that an initial priming phosphorylation site on the substrate interacts with the PP, enabling phosphorylation at secondary sites. In principle, an ideal priming site is phosphorylated rapidly, perhaps because it has a higher affinity for the kinase active site (due to a basic residue at the +3 position, for example). An ideal priming site must also have good affinity for the cyclin phosphate-binding pocket. We do not know how this affinity is determined, but it seems likely that sequences adjacent to the phosphorylation site interact with cyclin residues adjacent to the PP. We also do not know if the sequence context that promotes active site binding also promotes PP binding, so it is possible that certain poor CDK sites can bind effectively to the PP. On the other hand, what are the typical characteristics of a secondary site? First, we presume that these sites are suboptimal CDK sites, either because they lack the +3 K/R or even lack the +1 proline. Second, it seems likely that secondary sites must reside within some distance of the priming site on either the N- or C-terminal side, or both. These requirements remain unexplored, but our studies of various Cdc27 mutants, as well as our MS results, suggest that dividing sites into priming and secondary categories is not straightforward. Instead, we suspect that some sites can serve both as priming sites and as secondary sites, resulting in a complex variety of routes to full multisite phosphorylation.

Cks1 and other docking mechanisms are known to promote non-proline directed phosphorylation by CDKs[52,53]. We observed several non-consensus phosphorylation sites in our studies of multiple substrates (Fig. 6). The stoichiometry of phosphorylation of these sites, relative to consensus sites, was quite low in some cases, raising doubts about the importance of such sites. But some non-consensus sites, particularly in Bud6 and Spa2, were heavily modified and seem likely to play important regulatory roles. It will be critical in future studies to use mutagenesis to test the biological impact of these sites.

Our results are consistent with recent evidence that the phosphate-binding pocket of vertebrate cyclin B1 is required for normal progression through mitosis[54] and meiosis[55]. As in our studies, Heinzle et al.[54] demonstrated that mutation of the PP of human cyclin B1 decreased phosphorylation in vitro of non-consensus sites in several APC/C subunits. In light of these studies, and considering the high conservation of the phosphate pocket in the sequence of eukaryotic B-type cyclins, it is likely that the pocket is a highly conserved regulator of mitosis in most eukaryotes.

In summary, our evidence suggests that the phosphate pocket of B-type cyclins serves as a mechanism for promoting secondary phosphorylation of CDK substrates, similar to Cks1. This stepwise phosphorylation mechanism likely plays an important role in the temporal regulation of cell cycle events[7,8,12]. Past studies have demonstrated that the timing and extent of phosphorylation depend on phosphorylation site sequence context, kinase activity level, substrate localization, cyclin-specific docking motifs, Cks1 interactions, phosphatase specificity, and the positioning of phosphorylation sites. Our study adds another regulatory layer to this remarkably complex control system.

# Methods

## Strain and plasmid construction

All strains are derivatives of W303a (Supplementary Table 1). Strains used in the growth assay and DIC microscopy analysis in Fig. 1 were based on previously constructed *clb2-pp* strains[7]. All strains were constructed using PCR- and/or restriction digest-based homologous recombination with 50-bp flanking homology. All plasmids were constructed using Gibson assembly to recombine restriction-digested vectors with PCR products or synthetic DNA fragments (Integrated DNA Technologies, IDT) (Supplementary Table 2).

## Yeast proliferation

Yeast strains were grown in YPD (1% Yeast Extract, 2% Peptone, 2% Dextrose) at 30 °C unless specified otherwise. The optical density-based growth assay was carried out using a TECAN Spark 10 M plate reader. 100 µl of $OD_{600} = 0.05$ culture was added to the wells of a transparent, sterile, flatbottom, non-tissue culture treated 96-well plate (Falcon #351172), which was then sealed with Breathe-Easy sealing membrane (Diversified Biotech #BEM-1). Absorbance ($OD_{600}$) of each well was measured every 15 min for 24 h during a 30 °C incubation with shaking. Four replicates of each strain were analyzed. Growth rate was quantified by fitting absorbance data to a Logistic Growth Model on Prism (Graphpad). For analysis of growth on agar plates, 5 µl aliquots of 1:10 serial diluted culture from $OD_{600} = 0.1$ were spotted and incubated at the specified temperature. A final concentration of 15 µg/ml benomyl was included in the rich media for the microtubule destabilization experiments.

## Microscopy

Differential interference contrast (DIC) imaging was performed with a Nikon Ti2-E microscope and a CFI Plan Apochromat Lambda 100X/1.45 NA oil immersion objective lens. Images were captured with a Teledyne Photometrics Prime 95B 25 mm camera. Nikon NIS Elements software was used to drive stage movement and acquisition. Live imaging of yeast cells with mCherry-tagged Spc42 was conducted as described[33]. Imaging was performed using a Nikon Ti-E microscope equipped with a Yokogawa CSU-22 spinning disk confocal unit and a Teledyne Photometrics Evolve EMCCD camera. Imaging sessions typically lasted 1.5 h with 2 min between frames. For each time point and channel, z-stacks were acquired across a distance of 5 µm with 0.5 µm steps. The exposure time for the mCherry channel was set to be less than 40 ms for each z-slice. The imaging system was controlled using Nikon NIS Elements software, ensuring precise and accurate acquisition of the images.

Prior to fluorescence microscopy, yeast were grown for 24 h in synthetic complete media with 2% glucose (SD), with dilution to maintain $OD_{600} < 0.4$. For imaging, wells in a 96-well Glass Bottom Microwell Plate (MGB096-1-2-LG-L) were pre-treated with 10 mg/ml Concanavalin A for 1 h and washed with water. 100 µl cell culture was added and immobilized 40–60 min in a 30 °C incubator. Unattached cells were washed off with fresh media before imaging. To minimize autofluorescence, all media was filtered instead of autoclaved.

## Western blotting

Yeast strains expressed C-terminally Myc-tagged proteins either at the endogenous locus (securin) or at the *TRP1* locus under the control of the *GAL* promoter (Ndd1)[7]. For *GAL*-induced *NDD1* expression, galactose was added to a final concentration of 2% in YEP-raffinose medium. Cells were arrested by alpha factor (1.5 µg/ml) for 3 h at 30 °C and released from arrest by washing 3 times with fresh media. Samples (1 ml) were harvested at 20 min intervals after release. To block entry into the following cell cycle, alpha factor was added back to the culture 1 h after release, when most cells had budded. Yeast lysates were prepared by bead-beating cells for 2 min in 100 µl urea lysis buffer (20 mM Tris pH 7.4, 7 M urea, 2 M thiourea, 65 mM CHAPS, 10 mM DTT). Following SDS-PAGE of lysates, proteins were transferred to 0.45-micron nitrocellulose membrane (GE-Healthcare Life Sciences) at a constant voltage of 110 V for 70 min at 4 °C. Blots were incubated overnight at 4 °C with primary antibodies: either mouse anti-Myc (9B11 Cell Signaling Technology) diluted 1:5000, or GAPDH monoclonal antibody (GA1R Thermo) diluted 1:2500 in TBS-T containing 4% nonfat dry milk. After washing, blots were incubated at room temperature for 1 h with secondary antibody IRDye 800CW Donkey anti-Mouse IgG (926-32212 LI-COR), diluted 1:2000 in TBS-T. After washing, fluorescence signals on the blots were captured using an Odyssey Fc imager (LI-COR) at 800 nm for a duration of 10 min.

## Protein production

Cdk1 (Cdc28) was produced from a recombinant baculovirus encoding a GST-TEV-Cdk1 fusion. 800 ml of infected Sf9 cells were harvested and resuspended in buffer A (50 mM HEPES pH 7.5, 150 mM NaCl, 1 mM DTT, 10% glycerol) supplemented with 250 U of Benzonase and 1 cOmplete Protease Inhibitor Tablet. After sonication, the lysate was centrifuged at $256,630 \times g$ (Beckman Ti70 rotor, 50,000 rpm) for 1 h at 4 °C. Supernatant was loaded onto 300 µl Glutathione Sepharose 4B (Cytiva) for 1 h at 4 °C, washed 3 times with buffer B (50 mM HEPES pH 7.5, 1 M NaCl, 1 mM DTT, 10% Glycerol), and eluted with 4 ml buffer A supplemented with 10 mM reduced glutathione pH 8.0 (Sigma). The eluate was concentrated and subjected to gel filtration on an S200 column equilibrated in buffer C (50 mM HEPES pH 7.5, 150 mM NaCl, 0.5 mM TCEP, 10% Glycerol). Fractions containing GST-Cdk1 were pooled, concentrated, and snap frozen in liquid $N_2$.

For the production of Clb2, BL21(DE3) Star *E. coli*, transformed with a pET28 vector containing 6xHis-tagged wild-type or mutant Clb2 (aa 187-491), were grown in 1 liter of TB + Kanamycin at 37 °C to $OD_{600} = 0.6$ and induced with 1 mM IPTG at 16 °C overnight. Cells were frozen in liquid $N_2$, thawed, and resuspended in buffer A supplemented with 20 mM Imidazole, pH 8.0, 10 mg/ml DNaseI, 1 mM PMSF, 500 µg/ml Lyzosyme and 1 cOmplete protease inhibitor tablet. The resuspension was dounce-homogenized 3-5 times, passed through an 85 µm filter, and lysed in an LM10 (Microfluidics) microfluidizer, twice at 12,000 psi. The lysate was centrifuged at $185,510 \times g$ (Beckman Ti45 rotor, 40,000 rpm) for 30 min at 4 °C. Supernatant was incubated with Ni-NTA beads (QIAGEN #30230) for 1 h at 4 °C, washed 3 times with 10 bed volumes of buffer B and eluted with 4 ml buffer A supplemented with 200 mM Imidazole pH 8.0. The eluate was concentrated and subjected to gel filtration on an S200 column equilibrated in buffer C. Fractions containing Clb2 were pooled, concentrated, and snap frozen in liquid $N_2$.

Cak1 (*S. cerevisiae* CDK-Activating Kinase 1) was expressed from a recombinant baculovirus encoding a 6xHis-TEV-Cak1 fusion as described[45]. 800 ml infected Sf9 cells were harvested and resuspended in CAK buffer (50 mM HEPES pH 7.4, 250 mM NaCl, 2 mM TCEP, 5% Glycerol) supplemented with 15 mM Imidazole pH 8.0, 250 U of Benzonase, and 1 cOmplete Protease Inhibitor Tablet. After sonication, the lysate was centrifuged at $256,630 \times g$ (Beckman Ti70 rotor, 50,000 rpm) for 1 h at 4 °C. Supernatant was incubated with 2.5 ml Ni-NTA beads (QIAGEN #30230) for 1 h at 4 °C, washed 3 times with 25 ml CAK buffer, and eluted with 4 ml CAK buffer supplemented with 250 mM Imidazole pH 8.0. The eluate was concentrated and subjected to gel filtration on an S200 column equilibrated in CAK buffer. Fractions containing Cak1 were pooled, concentrated, and snap frozen in liquid $N_2$.

Active Clb2-Cdk1 complexes were assembled as previously described[45]. Briefly, in a 200 µl reaction mixture, 10.5 µM Cdk1 and 1.7 µM Cak1 were combined with 2.5 mM ATP and 10 mM $MgCl_2$ at 30 °C for 30 min. 10.5 µM Clb2 was added and incubated on ice for 60 min. The resulting complexes were desalted (Zeba Spin Desalting Column 7 K MWCO; Thermo Fisher) to remove ATP before snap freezing in liquid $N_2$. Concentrations of Cdk1 and Clb2 were quantified

against a BSA standard curve on an SDS-PAGE gel stained with SYPRO Red (Invitrogen).

Purification of wild-type and mutant (R33E, S82E, R102A) *S. cerevisiae* Cks1 was performed as described[20] with the following modifications. Briefly, BL21(DE3) Star cells transformed with a plasmid expressing *CKS1* from the T7 promoter were grown in 1 liter of TB + Ampicillin at 37 °C to $OD_{600} = 0.6$ and induced with 1 mM IPTG at 16 °C overnight. Cells were frozen in liquid $N_2$, thawed, sonicated in 10 ml phosphate-buffered saline pH 7.4, and clarified by centrifugation at $5680 \times g$ (Beckman Ti45 rotor, 7000 rpm). Supernatant was incubated in a boiling water bath for 5 min and centrifuged for 10 min at $185,510 \times g$ (Beckman Ti45 rotor, 40,000 rpm). Supernatant was supplemented with ammonium sulfate (1 g per 6.1 ml), incubated for 30 min at 4 °C, and centrifuged at $185,510 \times g$ for 10 min (Beckman Ti45 rotor, 40,000 rpm). The pellet was redissolved in 10 ml Tris-NaCl-EDTA buffer (50 mM Tris pH 7.5, 100 mM NaCl, 2 mM EDTA) and dialyzed against three changes of Tris-NaCl buffer (50 mM Tris pH 7.5, 100 mM NaCl). Dialysate was subjected to gel filtration on an S200 column in Tris-NaCl buffer, and fractions containing Cks1 were pooled and concentrated.

For the production of CDK substrates, codon-optimized synthetic DNA (IDT) was inserted into a pET28 expression vector by Gibson assembly, fused to DNA encoding an N-terminal 6xHis-Smt3 (SUMO) tag. BL21(DE3) Star cells transformed with plasmids expressing recombinant protein from the T7 promoter were grown in 2 liter of TB + Kanamycin at 37 °C to $OD_{600} = 0.6$ and induced with 1 mM of IPTG at 16 °C overnight. Cells were frozen in liquid $N_2$. Cell pellets were thawed and resuspended in native His lysis buffer (50 mM Tris pH 7.5, 300 mM NaCl, 20 mM Imidazole, 3 mM DTT, 5% Glycerol) supplemented with 10 mg/ml DNaseI, 250 U Benzonase, 1 cOmplete Protease Inhibitor Tablet, 500 µg/ml lysozyme and 1 mM PMSF. Suspended cells were lysed with an LM10 (Microfluidics) microfluidizer, twice at 12,000 psi. Lysate was then clarified by centrifugation at $185,510 \times g$ for 30 min (Beckman Ti45 rotor, 40,000 rpm). Supernatant was incubated with Ni-NTA beads (QIAGEN #30230) for 1 h at 16 °C, washed 3 times with 10 bed volumes of native His lysis buffer, and eluted with 4 ml 200 mM imidazole in the same buffer. The eluate was subjected to gel filtration on an S200 column equilibrated in 50 mM HEPES pH 7.5, 50 mM NaCl, 0.5 mM TCEP, 10% glycerol. Fractions containing appropriately sized proteins were pooled, concentrated, and snap frozen in liquid $N_2$.

## Kinase assays

APC/C immunoprecipitation was performed as previously described[37,56] with the following modifications. A *CDC16-TAP* yeast strain was grown to $OD_{600} = 1.0$. ~1.5 × 10^6 cells were pelleted into a 2 ml Sarstedt Mikro-Schraubröhre tube and snap frozen in liquid $N_2$. Thawed pellets were resuspended in 350 µl of APC/C lysis buffer (50 mM HEPES pH 7.8, 700 mM NaCl, 1 mM EDTA, 1 mM EGTA, 5% glycerol, 0.25% NP-0.4, 1 mM DTT, 1 mM PMSF, 1 cOmplete Protease Inhibitor tablet) and 1 ml of 0.5 mm diameter glass beads was added. Samples were bead-beaten by 2 pulses of 45 s with 5 min on ice between pulses. The bottom of the tube was punctured with a 25-gauge needle, and the lysate was separated from the glass beads by centrifugation at $500 \times g$ for 10 min (Beckman GH 3.8 rotor, 1500 rpm). Lysates were then clarified at $18,000 \times g$ (Eppendorf 5425r, 14,000 rpm) for 5 min at 4 °C. Samples were normalized to the same final protein concentration. For each reaction, 10 mg of protein was added to IgG-Dynabeads equilibrated in Kinase Bead Buffer (500 mM NaCl, 50 mM Tris pH 7.4, 50 mM NaF, 5 mM EGTA, 5 mM EDTA, 0.1% Triton X-100) and rotated for 2 h at 4 °C. Beads were washed 3 times with 150 µl Kinase Bead Buffer and 2 times with Low Salt Kinase Buffer (10 mM NaCl, 20 mM HEPES pH 7.4, 5 mM $MgCl_2$, 1 mM DTT). Samples were transferred to a new tube after the second wash. Beads were resuspended in Low Salt Kinase Buffer with 10 µM ATP, 2 µCi [γ–$^{32}$P]-ATP, and 250 nM okadaic acid. In parallel,

wild-type or PP mutant Clb2-Cdk1 was combined with equimolar mutant Cks1. Kinase reactions were initiated by the addition of Clb2-Cdk1-Cks1 complex to the resuspended beads carrying immunoprecipitated APC/C. Reactions were incubated at 25 °C on a rotating shaker (Eppendorf ThermoMixer F1.5) at 1000 rpm. Timepoints were taken by quenching the reaction with 100 mM EDTA. Beads were washed 3 times with 150 µl Kinase Bead Buffer plus 300 nM Okadaic Acid, and 2 times with Low Salt Kinase Buffer plus 300 nM Okadaic Acid. Samples were transferred to a new tube after the second wash. Reaction products were analyzed by SDS-PAGE on a 4–20% polyacrylamide gel. Autoradiography was performed with an Amersham Typhoon 5 Biomolecular Imager (GE Healthcare Life Sciences), and images were quantified using ImageQuant TL software (Amersham Biosciences).

Kinase reactions with purified Clb2-Cdk1-Cks1 were performed as previously described[7]. 50–200 nM Clb2-Cdk1, equimolar Cks1, 50 nCi [γ–$^{32}$P]-ATP (Hartmann Analytic), and 2.5–10 µM substrate were incubated in kinase buffer (25 mM HEPES pH 7.5, 100 mM NaCl, 10 mM $MgCl_2$, 1 mM DTT, 100 µM ATP) at 25 °C. To minimize background from Clb2 autophosphorylation, the Clb2-Cdk1-Cks1 complex was first incubated in kinase buffer, after which the substrate and [γ–$^{32}$P]-ATP were added to initiate radiolabeled substrate modification. Reactions were quenched at specific time points by the addition to SDS-PAGE sample buffer, and products were analyzed by SDS-PAGE with either 7.5% or 12.5% polyacrylamide and 50 µM Phos-tag (Wako Chemicals). Autoradiography was performed as described above.

## Mass spectrometry

Aliquots of kinase reactions were brought to 60 µl in 50 mM Ammonium Bicarbonate pH 8 and 25% acetonitrile. 100 mM Tris (2-carboxyethyl) phosphine (TCEP) was added to a final concentration of 5 mM and incubated at room temperature for 20 min. 500 mM iodoacetamide (IAA) was added to a final concentration of 10 mM and incubated in the dark for 20 min. 100 mM $CaCl_2$ was added to a final concentration of 1 mM, followed by the addition of 1 µg trypsin. After an overnight incubation at 37 °C, formic acid was added to 5%, and the mixture was diluted prior to concentration in a Speedvac to a 10 µl volume.

Trypsin-digested peptides were analyzed by online capillary nanoLC-MS/MS using a 25 cm reversed-phase column and a 10 cm precolumn fabricated in-house (50 µm inner diameter, packed with ReproSil-Gold C18-1.9 µm resin (Dr. Maisch GmbH)) that was equipped with a laser-pulled nanoelectrospray emitter tip. The precolumn used 3.0 µm packing (Dr. Maisch GmbH) and Kasil frit. Peptides were eluted at a flow rate of 100 nl/min using a linear gradient of 2–40% buffer D for 70 min (buffer D: 0.05% formic acid and 95% acetonitrile in water) in a Thermo Fisher Easy-nLC1200 nanoLC system. Peptides were ionized using a FLEX ion source (Thermo Fisher) using electrospray ionization into a Fusion Lumos Tribrid Orbitrap Mass Spectrometer (Thermo Fisher Scientific). Data were acquired in orbi-trap mode. Instrument method parameters were as follows: MS1 resolution, 120,000 at 200 m/z; scan range, 350 – 1600 m/z. The top 20 most abundant ions were subjected to higher energy collision-induced dissociation (HCD) with a normalized collision energy of 35%, activation q 0.25, and precursor isolation width 2 m/z. Dynamic exclusion was enabled with a repeat count of 1, a repeat duration of 30 s, and an exclusion duration of 20 s. RAW files were analyzed using PEAKS (Bioinformatics Solutions Inc.) with the following parameters: specific cleavage specificity at the C-terminal site of R and K, allowing for 1 missed cleavage, precursor mass tolerance of 15 ppm, and fragment ion mass tolerance of 0.5 Daltons. Methionine oxidation and phosphorylation of STY amino acids were set as variable modifications, and cysteine carbamidomethylation was set as a fixed modification. The *S. cerevisiae* proteome was used for the protein search, except for analysis of the Cdc27 mutant with S328 as the sole CDK site, where we used a yeast proteome in which the Cdc27 sequence was replaced with that of the mutant. Peptide hits were filtered using a 1% FDR.

## Statistics and reproducibility
Statistical analyses were performed using GraphPad Prism, as described in the figure legends. The reproducibility of all yeast and biochemical results was confirmed by the performance of at least three independent experiments.

## Reporting summary
Further information on research design is available in the Nature Portfolio Reporting Summary linked to this article.

## Data availability
The data supporting the findings of this study are contained within the paper and its supplementary files. The mass spectrometry proteomics data have been deposited to the ProteomeXchange Consortium via the PRIDE partner repository with the dataset identifier PXD062428. Source data are provided with this paper.

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

## Acknowledgements

We thank Chris Carlson, Chloe Ghent, Briana Marinoni, Emmy Delaney, Jonah Goodfried, Mart Loog, David Barford, Andreas Boland, and Thomas Mayer for helpful discussions, and Haley Gause and Kari Herrington for assistance with microscopy. This work was supported by a grant from the National Institute of General Medical Sciences (R35-GM118053 to D.O.M.). Fluorescence microscopy was performed at the UCSF Center of Advanced Light Microscopy, with support from a NIH S10 Shared Instrumentation grant (1S10OD017993-01A1). Mass spectrometry was performed at the Vincent J. Coates Proteomics/Mass Spectrometry Laboratory Core Facility, RRID:SCR_025852.

## Author contributions

H.Y.N. conceived the project, performed experiments, and analyzed results, with assistance from D.H.W. and A.N.A. and guidance from D.O.M. R.A.M. performed mass spectrometry analyses. H.Y.N. and D.O.M. wrote the paper with assistance from all other authors.

## Competing interests

The authors declare no competing interests.
