## [Transparent Peer Review file · Nature Communications]

Phosphate-binding pocket on cyclin B governs CDK substrate phosphorylation and mitotic timing

Corresponding Author: Dr David Morgan

Version 0:

Reviewer comments:

Reviewer #1

(Remarks to the Author)

In this study, Ng and colleagues build their previous report that mutating the putative phosphate-binding pocket (PP) in CLB2 causes defects in mitosis. They use a combination of genetics and proteomics to demonstrate that the PP mutant is required for the timely degradation of securin, and that this appears mostly to be a problem with APC/C activity, not to the SAC, and to reduced CDK activity dependent on Swe1. They use proteomic analysis to identify phosphorylation sites that change in the PP mutant and identify some APC/C subunits as hypophosphorylated, but whether and how this affects APC/C activity is unclear.

Overall, this study will be of interest to the cell cycle field by providing insights into the role of the PP in B-type cyclins in coordinating mitosis through enhancing substrate phosphorylation; therefore, in principle this study warrants publication in Nature Communications. As it stands, however, the study is incomplete and in my view could be improved by some relatively straightforward experiments.

- 1) Given the previous report using *Xemopus* Cyclin B1, it is possible that the PP mutant-CDK complex has lower activity because of reduced recognition by Mih1/Cdc25. The authors should blot for anti-phosphotyrosine 18 to exclude this possibility.
- 2) As mentioned in the Discussion, work from the Ly lab indicates that Cks1-binding enables Cyclin-CDK complexes to phosphorylate non-CDK consensus sites. There are indications that the PP also enables this increase in substrate recognition but this is not explicitly analysed here, though the authors should have the data to do this.
- 3) The authors have previously shown that securin degradation in yeast is regulated by its phosphorylation as well as by APC/C activity. Thus, an alternative explanation for the delay in securin degradation is a change in its CDK-dependent phosphorylation. This alternative should be addressed experimentally. The authors could usefully analyse the degradation of another APC/C substrate, e.g. CLB5, to support the conclusion that the PP mutant has altered APC/C activity.

Reviewer #2

(Remarks to the Author)

I have reviewed the mass spectrometry based phosphoproteomics analysis in the manuscript by Henry et al. Briefly the authors used SILAC quantification to compare the phosphoproteomes of wild-type and *clb2-pp* yeast cells to determine if the mitotic defects seen in the latter are due to defects in Cdk1 substrate phosphorylation. However, the design of the experiment is inadequate as there are no replicates and therefore no statistical tests are performed to identify significant hits or at least rank the hits by significance. The y-axis in the plots in Figure 3 should be $-\log_{10}p$ -value or $-\log_{10}q$ -value (volcano plots) rather than summed area. Additionally, wild-type and *clb2-pp* cultures were labelled with "light" or "heavy" isotopes respectively, however it is common for SILAC experiments to also reverse the labels to avoid biases and artifacts, but this was not done. In Supplementary Dataset 1, the "Phosphorylation Sites" sheet contains redundant sequences (PSMs), so the actual number of unique phosphosites is much lower and should be stated as well. Also, the "Log2 Normalized L/H" values appear all negative which suggests not properly normalized data; how the data was normalized is not clear. In the "SPTP Sites" sheet though the "Log2 Normalized L/H" values are both negative and positive with a media near zero. Why this discrepancy? If we assume that the \log_2 ratios are reliable, is there a statistical over-representation or enrichment of Cdk1 substrates in the reduced in *clb2-pp* phosphosites (more than expected by chance)? Very often phosphorylation differences

can be attributed to differences in total protein levels between conditions (e.g. a phosphorylation seen as up-regulated because the respective protein is upregulated). As the wild-type and clb2-pp yeast cells might have many baseline proteome differences, collection of matched total proteome data is warranted for normalization of the phosphorylation data. This is particularly important when multiple phosphosites on the same protein, all or most change in the same direction. I appreciate that the authors have done follow up in vitro experiments for some of the findings, however this does not negate the importance of a properly designed phosphoproteomics analysis that will result in a reliable "stand-alone" dataset. Lastly, the phosphoproteomics experiment should be noted as in vitro rather than in vivo.

Reviewer #3

(Remarks to the Author)

This manuscript reports the function of a recently discovered pocket that is generally present on B-type cyclins that docks phosphorylated residues (much in the same way as the cyclin-Cdk subunit Cks1 does). Similarly to Cks1, this pocket promotes multisite phosphorylation through the docking of already phosphorylated sites. The authors show the importance of this mechanism for targets such as the anaphase promoting complex the drive cells through mitosis in a timely manner. This is a very nice piece of work that is suitable for publication in Nature Communications following revision.

The main line of research that I would like to see strengthened relates the multisite phosphorylation of the specific substrates studied, such as Cdc16 or Cdc27. I want to know more about the phosphate binding pocket promotes multisite phosphorylation. Does the pocket bind phospho-serines or phospho-threonines in these cases? Which residues can serve as docking platforms? Is the processive or semi-processive phosphorylation proceed towards the N- or C-terminal direction of the protein, or both? Are there distance requirements? Obviously addressing all these questions is too much to ask, but, after reading the manuscript, I did feel wanting of a bit more information about the multisite phosphorylation mechanism promoted by the cyclin pocket.

Minor points: For Fig 1B, I suggest just calculating the time constant for the initial exponential growth rate. The logistic fit also involves reaching high ODs and slowing down cell growth and so is more complex.

For Fig 2C, are there replicates to support the statement "Here, western blotting of Ndd1 levels in cells released from a G1 arrest showed that the anaphase degradation of Ndd1, like that of securin, was delayed 10-20 minutes in the clb2-pp strain"

For Fig4&5: Do the authors have any idea why Ssk1 is dependent on Cks1 in the human cyclin experiment but not in the yeast cyclin experiment?

Version 1:

Reviewer comments:

Reviewer #1

(Remarks to the Author)

The authors have addressed my concerns and I recommend publication.

Reviewer #2

(Remarks to the Author)

I previously reviewed the mass spectrometry analysis in the manuscript by Henry et al. I appreciate the fact that the authors disclose their efforts to reproduce the original MS results, although these did not prove reproducible, likely due to the reasons they explain in the response letter.

The authors have now added new MS analysis of CDK phosphorylation sites in vitro with purified kinases using seven technical replicates of ten large-scale kinase reactions with selected substrates (immunopurified APC/C and polarisome) aiming to identify major phosphorylation sites in reactions with wild-type and PP mutant kinases. The authors used label-free quantitative MS to identify and quantify phosphorylation sites in these kinase reactions. Specifically, they selected sites for which chromatographic peak areas were obtained for both the modified and the corresponding unmodified peptides in four or more replicates to calculate the fraction of the phosphorylation site that is modified.

However, this approach for fraction calculation is not fully valid, primarily because phosphopeptides often exhibit different ionization efficiencies compared to their unmodified counterparts, which can significantly skew the calculated fractions. Additionally, phosphopeptides may also differ in their digestion efficiency, and since they elute at different time points during chromatography, they are subject to different levels of ion suppression or enhancement in the electrospray ionization (ESI) source. These factors collectively can lead to inaccurate estimation of phosphorylation occupancy if not properly accounted for. For these reasons the proteomics community has developed more sophisticated methods with phosphatase treatment and peptide labelling, such as the one developed by the Gygi lab <https://pubs.acs.org/doi/full/10.1021/acs.jproteome.7b00571>, for determining phosphorylation stoichiometry or occupancy. Therefore, the authors should utilize their quantitative phosphoproteomics data to directly compare phosphopeptide abundances between wild-type and mutant kinase reactions by performing appropriate statistical analyses, such as t-tests or

other differential expression methods. These comparisons should be filtered based on both p-value and effect size to identify significantly regulated phosphorylation events. As it stands, no statistical analysis is provided, despite the availability of multiple replicates.

The authors should also provide the full mass spectrometry output, including all quantitative values for every detected phosphopeptide across all samples (e.g., an Excel file with phosphopeptides as rows and individual samples as columns, containing LFQ or normalized intensities). Additionally, a clear explanation of the data normalization strategy should be included.

Currently, only phosphopeptides from the substrates are presented, but it is unclear how specific the kinase complex and substrate purifications were. Therefore, the full list of identified peptides/proteins, both phosphorylated and unmodified, should be reported to assess background and purification quality.

Importantly, the phosphorylation status of the substrates prior to kinase treatment is not described. This information is crucial for interpreting the results, as pre-existing phosphorylation could confound the assignment of kinase-dependent events.

Finally, the authors state that approximately 100 to 300 phosphopeptides were identified per substrate per replicate. It should be clarified whether these numbers refer to unique phosphopeptide sequences or include redundant PSMs.

Reviewer #3

(Remarks to the Author)

The authors have addressed my concerns.

Version 2:

Reviewer comments:

Reviewer #2

(Remarks to the Author)

The authors have addressed my comments, and I have no further concerns.

Response to Reviewers

Ng et al. [Nature Communications NCOMMS-24-17810-T]

We are grateful to the reviewers for their thoughtful comments and suggestions. We apologize for the lengthy delay in the submission of a revised version, which was the result of two factors: (1) due to technical difficulties in our studies of the effects of PP mutations on CDK substrate phosphorylation in vitro, we dedicated considerable effort to developing a new method for preparation of recombinant yeast Clb2-Cdk1-Cks1 complexes, allowing more refined control of reaction conditions and more robust results; and (2) to understand the effects of the phosphate pocket in more depth, we carried out a series of experiments to characterize patterns of substrate phosphorylation using mutagenesis and mass spectrometry. Due to these and other changes, the results previously found in the second half of the paper have been entirely replaced and expanded (previous Figures 3-5 are replaced with new Figures 3-6).

Changes in our kinase reactions should be explained in some depth. In our initial submission, we used Clb2-Cdk1 complexes purified from yeast in which a tagged Clb2 is overexpressed. In the months following our first submission, we discovered that new Clb2-Cdk1 preparations were not displaying the same PP-dependent defects that we observed with the preparations used for the work in the submitted manuscript. In general, the PP effects were much less pronounced, particularly with some substrates (Cdc16, Cdc27). After repeated attempts with new yeast-derived kinase preparations, we began to grow concerned that our early PP mutant kinase preparations were defective in some way, leading to variable effects on different substrates. Given that these kinases were purified from yeast in relatively low quantities, we were concerned that they might be contaminated or modified. To solve this problem, we developed a new method for large-scale heterologous expression of the kinase: yeast Cdk1 was produced in baculovirus-infected cells, phosphorylated with recombinant Cak1, and combined with Clb2 produced in bacteria. All proteins were well expressed and it was therefore possible to produce large-scale, homogeneous, and fully active kinase preparations, allowing us to carry out more well-defined explorations of kinase reaction conditions (variations in kinase and substrate concentrations, etc.). Using these conditions, we now show that the PP mutations have less dramatic but highly reproducible effects on multisite phosphorylation of four major substrates. We explored these effects in considerable detail by mutagenesis of sites in one substrate, Cdc27, and we also used quantitative mass spectrometry to identify phosphorylation sites and determine how they are affected by the PP mutation.

These and other changes are listed below in response to specific reviewer comments. Our responses to these comments are in blue font.

Reviewer #1:

Overall, this study will be of interest to the cell cycle field by providing insights into the role of the PP in B-type cyclins in coordinating mitosis through enhancing substrate phosphorylation; therefore, in principle this study warrants publication in Nature Communications. As it stands, however, the study is incomplete and in my view could be improved by some relatively straightforward experiments.

1) Given the previous report using Xenopus Cyclin B1, it is possible that the PP mutant-CDK complex has lower activity because of reduced recognition by Mih1/Cdc25. The authors should blot for anti-phosphotyrosine 18 to exclude this possibility.

This is an excellent point that we considered carefully. The main evidence against inhibitory phosphorylation of the PP complex in our previous manuscript is that this complex had the same Histone H1 kinase activity as the wild-type kinase. However, as mentioned above, we decided to rule out all PP-specific modifications by making new recombinant Cdk1 in baculovirus systems where inhibitory phosphorylation of Cdk1 does not occur. The new wild-type and PP mutant kinase preparations have identical activities toward numerous targets, such as the single-site Cdc27 substrates in Figure 4a.

2) As mentioned in the Discussion, work from the Ly lab indicates that Cks1-binding enables Cyclin-CDK complexes to phosphorylate non-CDK consensus sites. There are indications that the PP also enables this increase in substrate recognition but this is not explicitly analysed here, though the authors should have the data to do this.

Our new mass spectrometry data (Figure 6) address this comment in great detail: we now show that several non-consensus sites are phosphorylated in some substrates, and we show that phosphorylation at some of these sites is PP-dependent.

3) The authors have previously shown that securin degradation in yeast is regulated by its phosphorylation as well as by APC/C activity. Thus, an alternative explanation for the delay in securin degradation is a change in its CDK-dependent phosphorylation. This alternative should be addressed experimentally. The authors could usefully analyse the degradation of another APC/C substrate, e.g. CLB5, to support the conclusion that the PP mutant has altered APC/C activity.

This is a very insightful question. Our previous work (Holt et al. 2009) showed that phosphorylation of securin at two CDK sites inhibits securin ubiquitination by the APC/C, and our later work (Lu et al., 2014) showed that mutation of the two sites causes securin degradation to occur about 2 minutes earlier than usual. We predict that the PP mutation would cause a decrease in securin phosphorylation, resulting in slightly earlier degradation, not later. But we can't rule out unexpected scenarios in which the PP mutation somehow increases securin phosphorylation. To address this possibility, our western blotting results in Figure 2 include analysis of the timing of APC/C-dependent degradation of Ndd1, which is also delayed in the *clb2-pp* mutant.

Reviewer #2:

I have reviewed the mass spectrometry based phosphoproteomics analysis in the manuscript by Henry et al. Briefly the authors used SILAC quantification to compare the phosphoproteomes of wild-type and *clb2-pp* yeast cells to determine if the mitotic defects seen in the latter are due to defects in Cdk1 substrate phosphorylation. However, the design of the experiment is inadequate as there are no replicates and therefore no statistical tests are performed to identify significant hits or at least rank the hits by significance. The y-axis in the plots in Figure 3 should be $-\log_{10}p$ -value or $-\log_{10}q$ -value (volcano plots) rather than summed area. Additionally, wild-type and *clb2-pp* cultures were labelled with "light" or "heavy" isotopes respectively, however it is common for SILAC experiments to also reverse the labels to avoid biases and artifacts, but this was not done. In Supplementary Dataset 1, the "Phosphorylation Sites" sheet contains redundant sequences (PSMs), so the actual number of unique phosphosites is much lower and should be stated as well. Also, the "Log2 Normalized L/H" values appear all negative which suggests not properly normalized data; how the data was normalized is not clear. In the "SPTP Sites" sheet though the "Log2 Normalized L/H" values are both negative and positive with a

media near zero. Why this discrepancy? If we assume that the log₂ratios are reliable, is there a statistical over-representation or enrichment of Cdk1 substrates in the reduced in clb2-pp phosphosites (more than expected by chance)? Very often phosphorylation differences can be attributed to differences in total protein levels between conditions (e.g. a phosphorylation seen as up-regulated because the respective protein is upregulated). As the wild-type and clb2-pp yeast cells might have many baseline proteome differences, collection of matched total proteome data is warranted for normalization of the phosphorylation data. This is particularly important when multiple phosphosites on the same protein, all or most change in the same direction. I appreciate that the authors have done follow up in vitro experiments for some of the findings, however this does not negate the importance of a properly designed phosphoproteomics analysis that will result in a reliable “stand-alone” dataset. Lastly, the phosphoproteomics experiment should be noted as in vitro rather than in vivo.

We thank the reviewer for this very thorough analysis of the phosphoproteomics data in the first version of our manuscript. We decided to remove these results. In one attempt to repeat the experiment the results were not reproducible, and we are concerned that the mild phenotypes of the clb2-pp mutation (e.g. Figure 2) are accompanied by mild and noisy effects on large numbers of CDK substrates in the cell. It also seems likely that the effects will vary considerably at different cell cycle stages. A better approach might have been to analyze the triple clb1d clb2-pp clb3d mutant, or perhaps we should have arrested the cells in mitosis to enhance the detection of mitotic CDK substrates, as in our previous phosphoproteomics studies (Holt et al., 2009). We decided instead to remove this experiment and focus on a detailed biochemical analysis of CDK substrates chosen on the basis of their connection to the mitotic and budding phenotypes of the clb2-pp mutant.

We have now added mass spectrometry analysis of CDK phosphorylation sites in vitro with purified kinases (Supplementary Table 3, Figure 6). This experiment included seven technical replicates of ten large-scale kinase reactions, allowing identification of major phosphorylation sites in reactions with wild-type and PP mutant kinases.

Reviewer #3 (Remarks to the Author):

This manuscript reports the function of a recently discovered pocket that is generally present on B-type cyclins that docks phosphorylated residues (much in the same way as the cyclin-Cdk subunit Cks1 does). Similarly to Cks1, this pocket promotes multisite phosphorylation through the docking of already phosphorylated sites. The authors show the importance of this mechanism for targets such as the anaphase promoting complex the drive cells through mitosis in a timely manner. This is a very nice piece of work that is suitable for publication in Nature Communications following revision.

The main line of research that I would like to see strengthened relates the multisite phosphorylation of the specific substrates studied, such as Cdc16 or Cdc27. I want to know more about the phosphate binding pocket promotes multisite phosphorylation. Does the pocket bind phospho-serines or phospho-threonines in these cases? Which residues can serve as docking platforms? Is the processive or semi-processive phosphorylation proceed towards the N- or C-terminal direction of the protein, or both? Are there distance requirements? Obviously addressing all these questions is too much to ask, but, after reading the manuscript, I did feel wanting of a bit more information about the multisite phosphorylation mechanism promoted by the cyclin pocket.

These are excellent questions, and we are very interested in answering them. We attempted to address some of these questions in the revised paper through two main approaches: (1) mutagenesis of various combinations of the four CDK sites in one substrate, Cdc27; and (2) mass spectrometry analysis of in vitro phosphorylation sites in four substrates. The results with Cdc27 do not allow clear assignment of specific sites as priming sites or secondary sites; instead, it seems more likely that all four sites can act in either role. The mass spectrometry experiments identify the major sites of phosphorylation and reveal some sites that are PP-dependent, but here again they do not provide clear evidence for specific priming sites. Further understanding will require the identification or design of substrates in which the priming and secondary sites are clearly identified, so that we can rigorously address the interesting questions of N- versus C-terminal directions and distances.

Minor points: For Fig 1B, I suggest just calculating the time constant for the initial exponential growth rate. The logistic fit also involves reaching high ODs and slowing down cell growth and so is more complex.

Thank you for this suggestion. We did a simple calculation of the initial exponential growth rate and the results look very similar to those with the logistic fit, except that the R-values are not as good, especially for the slow-growing triple mutant. We therefore decided to stay with the logistic fit.

For Fig 2C, are there replicates to support the statement "Here, western blotting of Ndd1 levels in cells released from a G1 arrest showed that the anaphase degradation of Ndd1, like that of securin, was delayed 10-20 minutes in the *clb2-pp* strain"

Yes, we mention in the figure legend that the Ndd1 westerns are representative of two independent experiments.

For Fig 4&5: Do the authors have any idea why Ssk1 is dependent on Cks1 in the human cyclin experiment but not in the yeast cyclin experiment?

Our previous experiments with human Cdks have been removed so that the paper focuses entirely on the new more detailed studies with yeast kinases and substrates. Nevertheless, we can provide a likely explanation for the slight effect of Cks1 on Ssk1 phosphorylation in that previous experiment. Ssk1 does not contain threonine CDK sites and is not expected to interact with the phosphothreonine pocket of Cks1. However, our human kinase reactions contained purified cyclin B-Cdk1 at concentrations that were considerably higher than in the yeast reactions, and we suspect that these concentrations promoted nonspecific Cks1 interactions with phosphoserines on Ssk1. Note that the paper no longer includes Ssk1 and instead focuses on the four substrates most connected to the *clb2-pp* phenotypes.

Response to Reviewers

Ng et al. [Nature Communications NCOMMS-24-17810-A]

We thank the reviewer for the thorough analysis of our mass spectrometry results. Our responses are written below in blue font.

“The authors have now added new MS analysis of CDK phosphorylation sites *in vitro* with purified kinases using seven technical replicates of ten large-scale kinase reactions with selected substrates (immunopurified APC/C and polarisome) aiming to identify major phosphorylation sites in reactions with wild-type and PP mutant kinases. The authors used label-free quantitative MS to identify and quantify phosphorylation sites in these kinase reactions. Specifically, they selected sites for which chromatographic peak areas were obtained for both the modified and the corresponding unmodified peptides in four or more replicates to calculate the fraction of the phosphorylation site that is modified.”

There may be a misunderstanding about the composition of the kinase reactions used for our Phos-tag and MS analyses. We did perform one experiment with immunopurified APC/C (Fig 3a), but all subsequent experiments were performed with substrate fragments expressed at high levels in bacteria. The proteins were fused to an amino-terminal 6His-Smt3 tag, and metal affinity chromatography and gel filtration were used to generate highly purified substrates. Furthermore, as we described in our previous response to reviewers, we were concerned about contamination of our previous yeast kinase preparations and therefore developed an entirely new method of producing highly purified recombinant kinase from insect cells (Cdk1) and bacteria (Clb2 and Cks1). Thus, these are highly purified and well-defined reactions that do not contain significant amounts of contaminating proteins.

“However, this approach for fraction calculation is not fully valid, primarily because phosphopeptides often exhibit different ionization efficiencies compared to their unmodified counterparts, which can significantly skew the calculated fractions. Additionally, phosphopeptides may also differ in their digestion efficiency, and since they elute at different time points during chromatography, they are subject to different levels of ion suppression or enhancement in the electrospray ionization (ESI) source. These factors collectively can lead to inaccurate estimation of phosphorylation occupancy if not properly accounted for. For these reasons the proteomics community has developed more sophisticated methods with phosphatase treatment and peptide labelling, such as the one developed by the Gygi lab <https://pubs.acs.org/doi/full/10.1021/acs.jproteome.7b00571>, for determining phosphorylation stoichiometry or occupancy.

Therefore, the authors should utilize their quantitative phosphoproteomics data to directly compare phosphopeptide abundances between wild-type and mutant kinase reactions by performing appropriate statistical analyses, such as t-tests or other differential expression methods. These comparisons should be filtered based on both p-value and effect size to identify significantly regulated phosphorylation events. As it stands, no statistical analysis is provided, despite the availability of multiple replicates.”

We agree that the calculation of ‘fraction modified’ is not ideal for the reasons the reviewer describes. As suggested, we have now revised our results to simply compare the abundance of modified peptides in wild-type (WT) and mutant (PP) kinase reactions. To normalize the peak areas for substrate quantity, we used the total peak area for all peptides identified from the Smt3 protein, which is fused to the N-terminus of each substrate and therefore provides an accurate index of the amount of substrate in the reaction. In our revised Supplementary Table 4, the peak

area for each phosphorylated site is divided by the amount of Smt3 protein in that replicate to provide the normalized phosphopeptide area. Means of these normalized amounts are plotted in the revised Figure 6. Unpaired t-tests were used to compare WT and PP results. The revised results look very similar to the previous results, and most sites were not affected by the PP mutation. As before, the sites most clearly affected by the PP mutation are those detected in the WT reaction and not in the PP reaction.

In the Results text, we are quite cautious in our interpretation of these results. Indeed, we note that the PP mutation has little impact on most phosphorylation sites, and we devote the last paragraph of the Results to the question of why we see so little difference in many sites despite the effects we see on Phos-Tag gels.

“The authors should also provide the full mass spectrometry output, including all quantitative values for every detected phosphopeptide across all samples (e.g., an Excel file with phosphopeptides as rows and individual samples as columns, containing LFQ or normalized intensities). Additionally, a clear explanation of the data normalization strategy should be included.”

We added a new table, Supplementary Table 3, that lists all of the several thousand unique peptides identified in the substrates in our reactions. The quantity of each peptide (peak area) is provided for each of the seven replicates, when a chromatographic peak could be measured. We also provide the AScores for all phosphorylation sites where a phosphopeptide was detected in that replicate. This table illustrates a key point: the great majority of the phosphopeptides contain low-confidence phosphorylation site assignments that are unlikely to be correct. As noted in the text, all of the substrates contain large tryptic peptides containing a large number of potential phosphorylation sites, so that the same peptide was frequently identified in many different phosphorylated forms – most of which were low confidence and not selected for further study.

As before, the key results from our mass spectrometry analysis are found in a more refined table (previously Supplementary Table 3, now 4) that lists the small number of phosphorylation sites that passed our stringency tests. To assemble this table, we used the PTM Profiles function of the PEAKS software. This function selects high-confidence peptides using a specified threshold (2% ion intensity and AScore 15), and then aggregates the selected peptides on an exported table with individual rows for each phosphorylation site. The total peak area for each site is provided, along with the peak area of the corresponding unmodified peptide. Our Supplementary Table 4 is the result of gathering all PTM Profiles tables together and sorting according to kinase, substrates and sites. As described above, we have now added a column for the Smt3 peak area derived from the corresponding replicate. Division of the phosphopeptide peak area by the Smt3 peak area provides the phosphopeptide area that is normalized for substrate concentration.

“Currently, only phosphopeptides from the substrates are presented, but it is unclear how specific the kinase complex and substrate purifications were. Therefore, the full list of identified peptides/proteins, both phosphorylated and unmodified, should be reported to assess background and purification quality.”

As described above and in detail in our Methods section, the kinase and substrate preparations in our reactions are highly purified recombinant proteins that come mostly from bacteria, allowing us to create well-defined reactions that contained 200 nM kinase and 10 uM substrate. In our MS analysis, total peak areas indicate that the added substrate (and its Smt3 tag) was

generally present in ~50-100-fold excess over the three subunits of the kinase. None of the reaction components were prepared from yeast, and so there should be no yeast proteins in these reactions other than the substrate and kinase components. A small number of peptides were misidentified as yeast HSP peptides, presumably because these peptides have sequence homology to a bacterial protein co-purified with the substrate. There was also some carryover of substrates on the LC column, but these would not interfere with the analysis of substrate modifications. We are therefore confident that these reactions are well defined, and interference from other proteins is not significant.

“Importantly, the phosphorylation status of the substrates prior to kinase treatment is not described. This information is crucial for interpreting the results, as pre-existing phosphorylation could confound the assignment of kinase-dependent events.”

Substrates were produced in bacteria and do not contain phosphorylated residues. In one of our early pilot experiments with Cdc27, we confirmed that no residues are phosphorylated in kinase reactions lacking ATP.

“Finally, the authors state that approximately 100 to 300 phosphopeptides were identified per substrate per replicate. It should be clarified whether these numbers refer to unique phosphopeptide sequences or include redundant PSMs.”

These are unique phosphopeptides. However, we decided to remove the sentence with these numbers because the numbers are misleading: most of these phosphopeptides are not of sufficient quality to allow accurate phosphorylation site assignment, as illustrated by the new Supplementary Table 3.